# A SAM-I riboswitch with the ability to sense and respond to uncharged initiator tRNA

Dong-Jie Tang [1,7], Xinyu Du[2,3,7], Qiang Shi[4,7], Jian-Ling Zhang [1,6,7], Yuan-Ping He [1], Yan-Miao Chen [1], Zhenhua Ming[1], Dan Wang [1], Wan-Ying Zhong [1], Yu-Wei Liang[1], Jin-Yang Liu [1], Jian-Ming Huang[3], Yun-Shi Zhong[4], Shi-Qi An[5], Hongzhou Gu [2,3,4✉] & Ji-Liang Tang [1✉]

All known riboswitches use their aptamer to senese one metabolite signal and their expression platform to regulate gene expression. Here, we characterize a SAM-I riboswitch (SAM-I$_{Xcc}$) from the *Xanthomonas campestris* that regulates methionine synthesis via the *met* operon. In vitro and in vivo experiments show that SAM-I$_{Xcc}$ controls the *met* operon primarily at the translational level in response to cellular S-adenosylmethionine (SAM) levels. Biochemical and genetic data demonstrate that SAM-I$_{Xcc}$ expression platform not only can repress gene expression in response to SAM binding to SAM-I$_{Xcc}$ aptamer but also can sense and bind uncharged initiator Met tRNA, resulting in the sequestering of the anti-Shine-Dalgarno (SD) sequence and freeing the SD for translation initiation. These findings identify a SAM-I riboswitch with a dual functioning expression platform that regulates methionine synthesis through a previously unrecognized mechanism and discover a natural tRNA-sensing RNA element. This SAM-I riboswitch appears to be highly conserved in *Xanthomonas* species.

[1] State Key Laboratory for Conservation and Utilization of Subtropical Agro-bioresources, College of Life Science and Technology, Guangxi University, Guangxi, China. [2] Shanghai Key Laboratory of Medical Epigenetics, the International Co-laboratory of Medical Epigenetics and Metabolism, Ministry of Science and Technology, Institutes of Biomedical Sciences, Fudan University, Shanghai, China. [3] Center for Medical Research and Innovation, Shanghai Pudong Hospital, Fudan University Pudong Medical Center, Shanghai, China. [4] Endoscopy Center, Zhongshan Hospital of Fudan University, Shanghai, China. [5] National Biofilms Innovation Centre (NBIC), Biological Sciences, University of Southampton, University Road, Southampton SO17 1BJ, UK. [6] Present address: School of Public Health, Zunyi Medical University, 563000 Zunyi, Guizhou, China. [7] These authors contributed equally: Dong-Jie Tang, Xinyu Du, Qiang Shi, Jian-Ling Zhang. ✉email: hongzhou.gu@fudan.edu.cn; jltang@gxu.eu.cn

Riboswitches are cis-acting regulatory mRNA elements that are usually located in the 5′ untranslated region (5′UTR) of a messenger RNA (mRNA) and control gene expression by directly sensing small molecules[1–5]. Since their first discovery in 2002[1,6,7], riboswitches have become recognized as important and widespread regulators of genes involved in many bacterial cellular processes[8–12]. Currently, almost 40 distinct classes of riboswitch have been identified[13]. A riboswitch typically consists of two functional domains called the aptamer and the expression platform. The aptamer directly binds to a specific small molecule, and the expression platform undergoes structural changes in response to the stabilization of the aptamer structure and then regulates gene expression[8–12]. The majority of riboswitches have been shown to specifically sense and bind small molecules that include purines, amino acids, vitamins, co-factors, second messengers, and transfer RNA (tRNA)[8–13]. In this way, riboswitches can control a wide spectrum of cellular processes including vitamin metabolism, nucleotide and amino acid biosynthesis, and sulfur metabolism[8–15].

Methionine (Met) is a unique proteinogenic amino acid which plays acritical role in the initiation of translation and the precursor of the principal cellular methyl group donor S-adenosylmethionine (SAM)[16]. It has been shown in Gram-positive bacteria that the key regulators of Met biosynthesis are the SAM-I[14,15,17,18] and T-box[8,11,12,19–22] riboswitches. The SAM-I (also called S-box) are a class of riboswitch that regulate gene expression in response to SAM binding. In addition to modulating Met biosynthesis, SAM-I is also involved in cysteine biosynthesis, sulfur metabolism and SAM biosynthesis[14,15,17,18]. In contrast, members of the T-box class of riboswitch monitor the aminoacylation status of specific tRNAs to induce the expression of regulated downstream gene(s), involved in the biosynthesis of Met and other amino acids[8,11,12,19–22]. Interestingly, SAM-I and T-box riboswitches use opposite strategies to control Met biosynthesis: SAM-I uses a negative feedback mechanism to turn off Met biosynthesis in response to increasing SAM concentration[14,15,17,18], while T-box uses a positive feedback mechanism to turn on Met biosynthesis in response to the accumulation of uncharged Met-tRNA[8,11,12,19–22]. The regulation of Met de novo biosynthesis in Gram-negative bacteria was discovered to be controlled by regulatory proteins[23]. In the model organism Escherichia coli, MetR and MetJ have been demonstrated to be specifically involved in the control of Met biosynthesis. The MetR protein has been shown to act as a transcriptional activator which uses homocysteine as an inducer[23]. In contrast, the MetJ has been demonstrated to function as a transcriptional repressor using SAM as co-repressor[23]. This system of regulation in E. coli appears to be conserved in a high proportion of Gram-negative bacteria including the Xanthomonas genus[24]. Although potential riboswitches involved in the regulation of Met biosynthesis genes have been proposed in Gram-negative bacteria[3,24], none of them has been functionally characterized. T-box riboswitches have long been thought to exist primarily in Gram-positive bacteria[8,11,12,19–22,25].

Recent work examining the regulation of Met biosynthesis in the phytopathogen Xanthomonas campestris pv. campestris (hereafter Xcc) provided functional evidence of a Gram-negative bacterium utilizing a 5′UTR region to control the expression of the genes involved in the generation of Met[26]. As well as being a plant pathogen of global concern, Xcc is a model organism for molecular studies of plant-microbe interactions[27]. The mechanism by which this 5′UTR region exerts its regulatory action is incompletely understood. Here, we provide evidence demonstrating that this 5′UTR region from Xcc encodes a functional SAM-I riboswitch. Genetic and biochemical studies confirm that SAM-I$_{Xcc}$ modulates met operon expression primarily at the

translational level. Further analysis reveals that the SAM-I riboswitch from Xcc displays previously uncharacterized regulatory actions associated with the SAM-I class where the expression platform shows dual functionality. We demonstrate that the expression platform of SAM-I$_{Xcc}$ is involved in feedback regulation of the met operon in response to Met availability. In addition, we demonstrate that the SAM-I$_{Xcc}$ expression platform also functions as a sensor monitoring uncharged initiator Met-tRNA. The findings describe a structurally typical SAM-I riboswitch from Xcc with a previously uncharacterized mode of action. SAM-I$_{Xcc}$ appears to be broadly distributed in Gram-negative Xanthomonas species bacteria and its expression platform represents a type of natural tRNA-sensing RNA elements.

## Results

**SAM-I$_{Xcc}$ controls the met operon primarily at translation.** Our previous work demonstrated that the met operon is essential for Met de novo biosynthesis in Xcc strain 8004 and that a 5′UTR tightly regulates the operon in response to cellular levels of Met[26]. Further sequence analysis of the 5′UTR revealed a putative 200-nucleotide (nt) SAM-I-like riboswitch (designated going forward as SAM-I$_{Xcc}$) (Supplementary Fig. 1), similar to the SAM-I predicted previously[28]. The met operon consists of three genes, i.e., XC1251 (metA), XC1252 (metB), and XC1253 (hom), which encod a homoserine O-succinyltransferase, a cystathionine γ-synthase and a homoserine dehydrogenase, respectively (Fig. 1a). In addition to XC1251, XC1889 in the genome of Xcc strain 8004 also encodes a homoserine O-succinyltransferase[29], whose promoter region does not contain sequences similar to SAM-I$_{Xcc}$, suggesting that the expression of the two homoserine O-succinyltransferase-encoding genes may be regulated by different modes. The predicted aptamer of SAM-I$_{Xcc}$ displayed a 52% sequence similarity to the aptamer of yitJ SAM-I from Bacillus subtilis[14]. SAM-I$_{Xcc}$ does not contain an Rho-independent transcription terminator. Given that all of the functionally characterized SAM-I riboswitches employ the Rho-independent terminator to control gene expression at the transcriptional level[14–16,18], we presume that SAM-I$_{Xcc}$ may use the translation attenuation mechanism to regulate gene expression, although the possibility of using the Rho-dependent transcription termination mechanism can not be excluded.

To examine the potential role of SAM-I$_{Xcc}$ in gene regulation in reaction to cellular levels of SAM, we used several reporter constructs carrying SAM-I$_{Xcc}$ fused to the gusA gene from E. coli (Supplementary Fig. 2). Two SAM-I$_{Xcc}$-gusA fusion reporters were created to monitor transcriptional (pWT-SD$^+$) and translational (pWT-SD$^-$) activity and introduced into the Xcc 8004 wild-type strain (see Methods; Fig. 1b). It is known that bacteria can take up SAM directly by a SAM-specific transporter[30,31]. The growth of Xcc met operon inactivation mutant 1201PK2 (Supplementary Table 1), which is unable to synthesize Met and SAM[26], could be restored in the minimal medium MMX with addition of SAM (Supplementary Fig. 3), suggesting the presence of SAM transporter in Xcc. For the reporter strain Xcc 8004/pWT-SD$^+$, the GUS activity was repressed by ~24% when grown in the medium supplemented with 300 μM SAM relative to medium with no SAM supplementation (Fig. 1b). However, GUS activity observed for the Xcc 8004/pWT-SD$^-$ strain was repressed by ~98% in the medium supplemented SAM relative to medium with no SAM supplementation (Fig. 1b). In addition, both reporter strains showed negligible change in GUS activity when the medium was supplemented with an alternative amino acid, glycine, at a concentration of 300 μM (Fig. 1b).The data indicate that SAM-I$_{Xcc}$ is specifically responsive to the cellular levels of SAM and

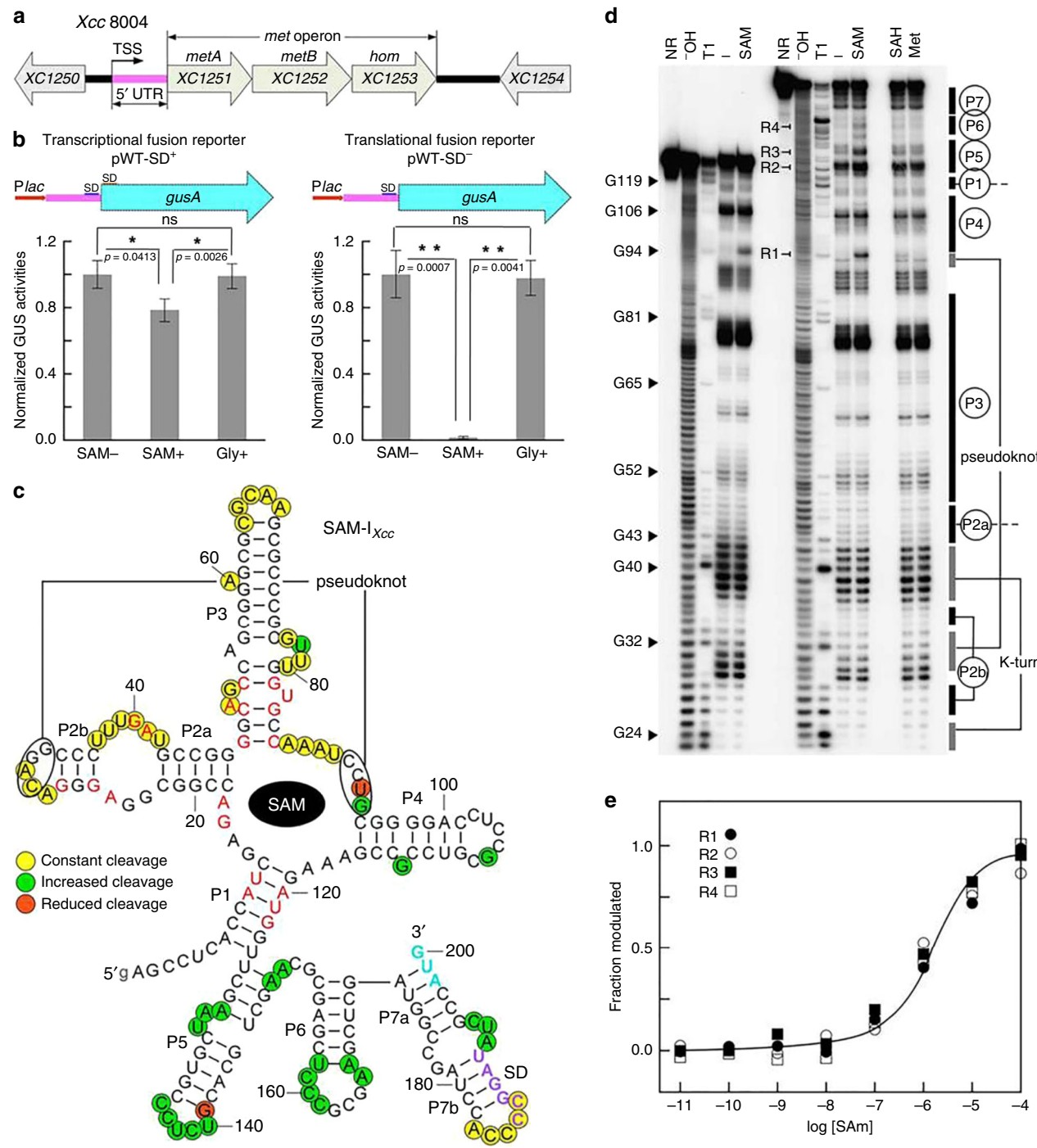

controls gene expression primarily at the translational level. In addition, the GUS activity of the transcriptional fusion reporter strain (8004/pWT-SD+) shows a small but statistically significant reduction upon addition of SAM, suggesting that SAM-I$_{Xcc}$ may also modulate gene expression weakly at the transcriptional level. However, we cannot rule out that this reduction of GUS activity may be caused by an influence on mRNA stability induced by the binding of SAM to the aptamer or that this effect may be due to indirect effects of SAM on transcription in general.

To test whether the three consecutive hairpin structures (P5, P6, and P7) formed in the expression platform upon SAM binding to the aptamer (Fig.1c) is involved in the reduction of GUS activity of the transcriptional fusion reporter strain in response to SAM addition, *gusA* transcriptional fusion reporters

carrying a series of full-length or truncated expression platforms from SAM-I$_{Xcc}$ were constructed (Supplementary Fig. 4) and their GUS activities were determined in the presence and absence of SAM. The result showed that the three hairpins together and the combination of P5 and P6 or P6 and P7 hairpins can reduce the GUS activity although the efficiency is much lower than that of the *trp* terminator (Supplementary Fig. 4). It is possible that these hairpins act as a transcription attenuator or serve as an RNase-binding target to recruit RNase which then degrades the mRNA, or cause other indirect effects.

To assess ligand binding in vitro, the predicted 200-nt SAM-I$_{Xcc}$ (Fig. 1c) was subjected to in-line probing analysis as described in Methods[32,33]. Due to its high G+C content, ligand-induced changes in spontaneous RNA cleavage of SAM-I$_{Xcc}$ were only

**Fig. 1 Identification of SAM-I$_{Xcc}$ riboswitch. a** The genetic organization of the *met* operon (*XC1251-1253*) (*XC1251*, *metA*, encoding homoserine O-succinyltransferase; *XC1252*, *metB*, encoding cystathionine γ-synthase; *XC1253*, *hom*, encoding homoserine dehydrogenase) locus and the 5′UTR location (magenta) where SAM-I$_{Xcc}$ is positioned in Xcc strain 8004's genome. Arrows indicate the transcription orientation of genes. TSS, the transcription start site of the *met* operon. **b** Fusion reporter constructs and their GUS activity assay. SD, Shine-Dalgarno sequence. Plot depicts the level of the fused *gusA* reporter gene expression in MMX medium alone (SAM−), and MMX with 300 µM SAM (SAM+) or 300 µM glycine (Gly+). GUS activities produced by strain 8004/pWT-SD$^+$ were 80.6 (±8.7) for SAM−, 61.2 (±6.5) for SAM+, and 78.9 (±9.1) for Gly+; produced by strain 8004/pWT-SD$^−$ were 4.13 (±0.38) for SAM-, 0.08 (±0.007) for SAM+, and 3.98 (±0.41) for Gly+. Data are presented as mean values ± SD from three biologically independent samples. Asterisks represent the significant difference at $P < 0.05$ (one asterisk) or $P < 0.01$ (two asterisks) by Student's two-tailed *t*-test. ns, not significant. **c** Structural modulation of SAM-I$_{Xcc}$ from the 5′UTR. Conserved nucleotides are indicated by red in SAM-I consensus model. Other highlighted nucleotides indicate locations of spontaneous cleavage upon addition of SAM, which were mapped using the in-line probing data (see Supplementary Figs. 5-8 and the next panel). **d** In-line probing analysis of the aptamer (left) and full-length (right) of SAM-I$_{Xcc}$. NR, $^−$OH, and T1 designate no reaction, partial digestion with alkali and RNase T1 (G-specific cleavage), respectively. Samples in the remaining lanes were incubated in absence (−) or presence of 100 µM SAM, 1 mM SAH, or 1 mM Met. R1-4 identifies major spontaneous RNA cleavage changes brought about by SAM. **e** Plot of the fraction of SAM-I$_{Xcc}$ bound to SAM versus the logarithm of SAM concentration (M) as inferred from the modulation of spontaneous cleavage products from in-line probing (panel d and Supplementary Fig. 9). $N = 4$ bands (R1-4) examined over 3 independent in-line probing experiments. Source data are provided as a Source Data file.

seen when the reaction temperature was above 37 °C (Supplementary Fig. 5). Upon PAGE separation, the pattern of RNA cleavage products in the presence of SAM (Fig. 1d, Supplementary Figs. 6-9) was consistent with our predicted secondary structure model (Fig. 1c). More than 20 linkages (Fig. 1c) exhibit increased strand scission in the concentration of 100 µM SAM, indicating that these nucleotides are structurally exposed during the SAM-binding induced reorganization. Conversely, no structural modulation was evident upon the introduction of Met and *S*-adenosyl-L-homocysteine (SAH) up to a concentration of 1 mM, revealing great molecular discrimination of SAM-I$_{Xcc}$. In-line probing using a range of SAM concentrations (Supplementary Fig. 9) suggests a dissociation constant ($K_D$) in one-to-one binding of ~2 µM for the 200-nt SAM-I$_{Xcc}$ (Fig. 1e), which is an order of magnitude weaker than that of the 251-nt *yitJ* SAM-I RNA ($K_D$ ~200 nM)[14]. Since modulation could only be seen in the in-line probing performed at a higher temperature (37 °C) than that generally used (22 °C) (Supplementary Fig. 5), it is reasonable to believe that the low binding affinity of SAM-I$_{Xcc}$ could be attributed to this.

The SAM-I[18] and SAM-IV[34] riboswitches are known to possess similar SAM-binding core and can be distinguished by different architectural elements and nucleotide conservation patterns in many places[34]. The overall architecture of the SAM-I$_{Xcc}$ aptamer corresponds to SAM-I aptamer rather than SAM-IV aptamer, including a P4 hairpin in the core, a lack of an additional pseudoknot, a kink-turn in the P2 stem and uridine residue at position 121 (Fig. 1c, Supplementary Fig. 10). Importantly, mutations of known conserved SAM-binding sites within SAM-I aptamer in SAM-I$_{Xcc}$ resulted in the loss of SAM-responsive regulation in the SAM-I$_{Xcc}$-*gusA* fusion reporter strain (Supplementary Fig. 11).

**SAM-I$_{Xcc}$ SD sequence is sequestered in SAM-unbound state**. It is clear that SAM-I$_{Xcc}$ is responsive to the cellular levels of SAM and appears to play an important role in inhibiting translation (Fig. 1). The potential mechanism by which SAM-I$_{Xcc}$ inhibits translation likely involves SAM-mediated structural rearrangements that inhibit translation initiation. The in-line probing experiments revealed structural transitions seen by SAM-I$_{Xcc}$ in the presence and absence of SAM (Fig. 2a; Supplementary Figs. 6-9). The SAM-I$_{Xcc}$ model developed from the in-line probing data suggests that in the SAM-bound state, the two important RNA elements involved in translation initiation (the SD sequence and the start codon AUG) are sequestered within the expression platform's hairpin structure (Fig. 2a; Supplementary Fig. 10). In the SAM-unbound state, SAM-I$_{Xcc}$ seems to fold

in an alternative structure where the start codon AUG is exposed, but the SD sequence is still sequestered by the expression platform's short hairpin, implying that continued translation repression may occur (Fig. 2a). We attempted to validate this mechanism by constructing several SAM-I$_{Xcc}$ mutants that carry disruptive changes in the expression platform in GUS reporter constructs (see Methods, Fig. 2b). As expected, the inhibition by SAM was completely abolished in the translational reporter strain carrying the construct with 9-nucleotide changes in the anti-AUG and anti-SD sequences (M1 + M2), which in theory disrupts the sequestration hairpin and releases both the SD and AUG (Fig. 2c). The mutant (M2) exposing the SD only exhibited about 51% GUS reporter activity (Fig. 2c), indicating partial translation inhibition in the presence of SAM and suggesting the independent sequestration of AUG. The construct (M1) exposing the start codon AUG showed about 50% GUS reporter activity (Fig. 2c), suggesting the independent sequestration of SD. These results are consistent with the predicted structural model of SAM-I$_{Xcc}$ in the SAM-bound state but inconsistent with the model in the SAM-free state. As shown in Fig. 2a, the SD sequence is still sequestered and thus translation repression should occur in the SAM-free state. However, in absence of SAM, the GUS activities produced by the reporter strains carrying the wild-type SAM-I$_{Xcc}$ construct (pWT-SD$^−$) or the mutant construct (M2) exposing the SD are very similar (Fig. 2c), suggesting that the wild-type SAM-I$_{Xcc}$ is fully switched on and both the SD and AUG are accessible in absence of SAM. The simplest explanation for this is that the anti-SD hairpin may not exist in the SAM-free state. However, the RNase H cleavage experiments revealed that in absence of SAM the anti-SD hairpin exists (Fig. 3b, and Supplementary Fig. 12). These results support the predicted SAM-I$_{Xcc}$ structure model and suggest that an additional factor(s) may contribute to the accessibility of SD when the cellular SAM is deficient.

**SAM-I$_{Xcc}$ expression platform can bind uncharged tRNA$^{fMet}$**. In addition to SAM-I, the T-box riboswitches found in various Gram-positive bacteria have been shown to regulate Met biosynthesis in response to the accumulation of uncharged Met-tRNA[8,35,36]. The homoserine O-acetyltransferase (encoded by *XC1251*) is connected to Xcc Met metabolism pathway and, by extension, to uncharged/charged Met-tRNA (Supplementary Fig. 13). Therefore, it is conceivable that the additional factor that might contribute to SAM-I$_{Xcc}$ SD accessibility is Met-tRNA. If SAM-I$_{Xcc}$ was to interact with Met-tRNA, it would likely use a different mechanism given that it does not have features

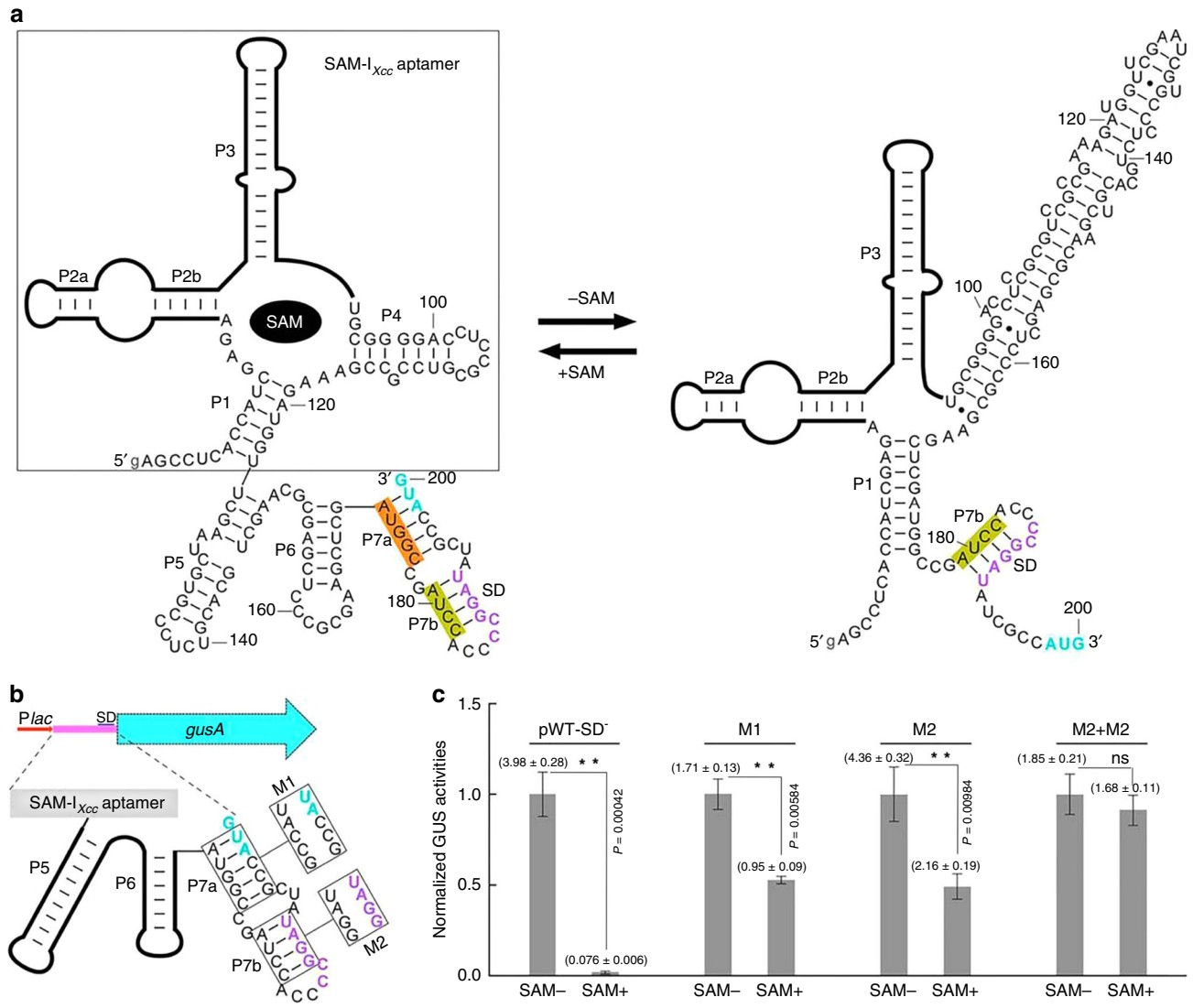

**Fig. 2 SAM-I$_{Xcc}$ regulates translation in response to SAM. a** The two structural states for SAM-I$_{Xcc}$ riboswitch in the presence and absence of SAM. The structural model was inferred using the data in Supplementary Figs. 6-9. Nucleotides in purple and cyan correspond to the SD sequence and the start codon, respectively. Nucleotides with yellow and orange shading refer to the anti-SD and the anti-AUG sequences, respectively. Sequences boxed belong to the aptamer region of the riboswitch. **b** Mutant constructs examined for the *gusA* gene expression in the translational fusion reporter. **c** Plot of the GUS activity from various reporter constructs in response to SAM. Data are presented as mean values ± SD from three biologically independent samples. Asterisks refer to the significant difference between SAM− and SAM+ of the same reporter strain, at $P < 0.01$ by Student's two-tailed *t*-test. ns, not significant. The values shown in the parentheses above the columns are the absolute GUS activity. Source data are provided as a Source Data file.

including the "specifier sequence" of a typical T-box riboswitch[8] (Supplementary Fig. 14).

To examine whether SAM-I$_{Xcc}$ can interact with Met-tRNA, we employed the electrophoretic mobility shift assay (EMSA). For these experiments, SAM-I$_{Xcc}$ and uncharged Met-tRNA were examined in vitro (see Methods, Fig. 4a-e). SAM-I$_{Xcc}$ was incubated with the three Met-tRNAs encoded in the genome of *Xcc* strain 8004 (Supplementary Fig. 15), i.e., the initiator Met-tRNA (tRNA$^{fMet}$, encoded by *XC4339*), the elongator Met-tRNA tRNA$^{Met1}$ (encoded by *XC4335*) or the tRNA$^{Met2}$ (encoded by *XC4381*), respectively. Bands that correspond to the complex between tRNA$^{fMet}$-SAM-I$_{Xcc}$ were observed but no complex between SAM-I$_{Xcc}$ and elongator Met-tRNA tRNA$^{Met1}$ or tRNA$^{Met2}$ was seen (Fig. 4c), suggesting that SAM-I$_{Xcc}$ can selectively bind with tRNA$^{fMet}$. The specificity of the interaction between SAM-I$_{Xcc}$ and tRNA$^{fMet}$ was confirmed by competitive EMSA (Supplementary Fig. 16). In addition to the three Met-tRNAs, the genome of *Xcc* strain 8004 was predicted to encode 51

other tRNAs[29] (Supplementary Table 3), nineteen of which were subjected to EMSA to examine whether they can interact with SAM-I$_{Xcc}$. The result showed that none of them could bind with SAM-I$_{Xcc}$ (Supplementary Fig. 17), further supporting the specificity of SAM-I$_{Xcc}$-tRNA$^{fMet}$ interaction.

In addition, the aptamer domain and the expression platform domain of SAM-I$_{Xcc}$ were isolated and incubated independently with tRNA$^{fMet}$ (see Methods). A band indicating a complex between the expression platform domain and tRNA$^{fMet}$ was detected but no band was seen when the aptamer domain and tRNA$^{fMet}$ were incubated together (Fig. 4c). The data indicate that tRNA$^{fMet}$ binds directly to the expression platform of SAM-I$_{Xcc}$. Additional assays were carried out to examine the interaction of the expression platform with variant tRNA$^{fMet}$ constructs (Fig. 4a). These variants mimicked charged tRNA$^{fMet}$ (M$_{3'+C}$), had a deletion of the 3'CCA (M$_{del}$), changed the 3'CCA to GGU, or carried a mutation of the anticodon (M$_{anti}$) in tRNA$^{fMet}$. Severely weakened binding was seen between the

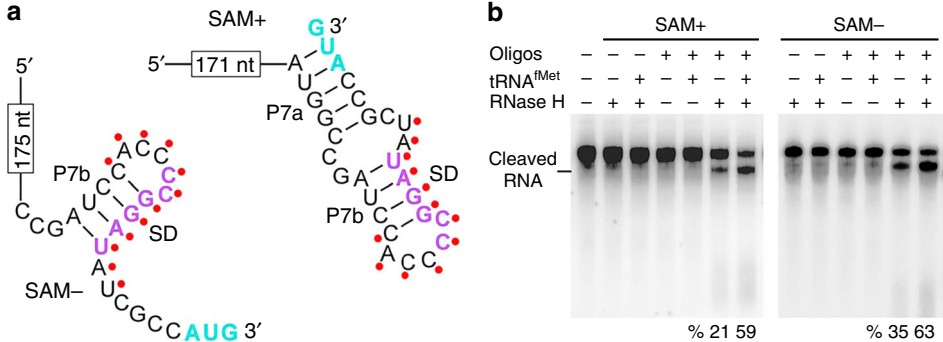

**Fig. 3 The binding of SAM-I$_{Xcc}$ with tRNA$^{fMet}$ leads to structural changes and release of SD sequence. a** The secondary structure of the 3' region of SAM-I$_{Xcc}$ in the presence (+, right) and absence (−, left) of SAM. Nucleotides highlighted by red dots were chosen for the hybridization with a synthetic complementary DNA oligo. **b** Observation of the SD release upon tRNA$^{fMet}$ binding to SAM-I$_{Xcc}$. Conformational changes in the SD region were monitored by using the antisense DNA oligo and RNase H cleavage analysis. Cleaved DIG-labeled SAM-I$_{Xcc}$ RNA was pointed out, and the cleavage yield was calculated. Source data are provided as a Source Data file.

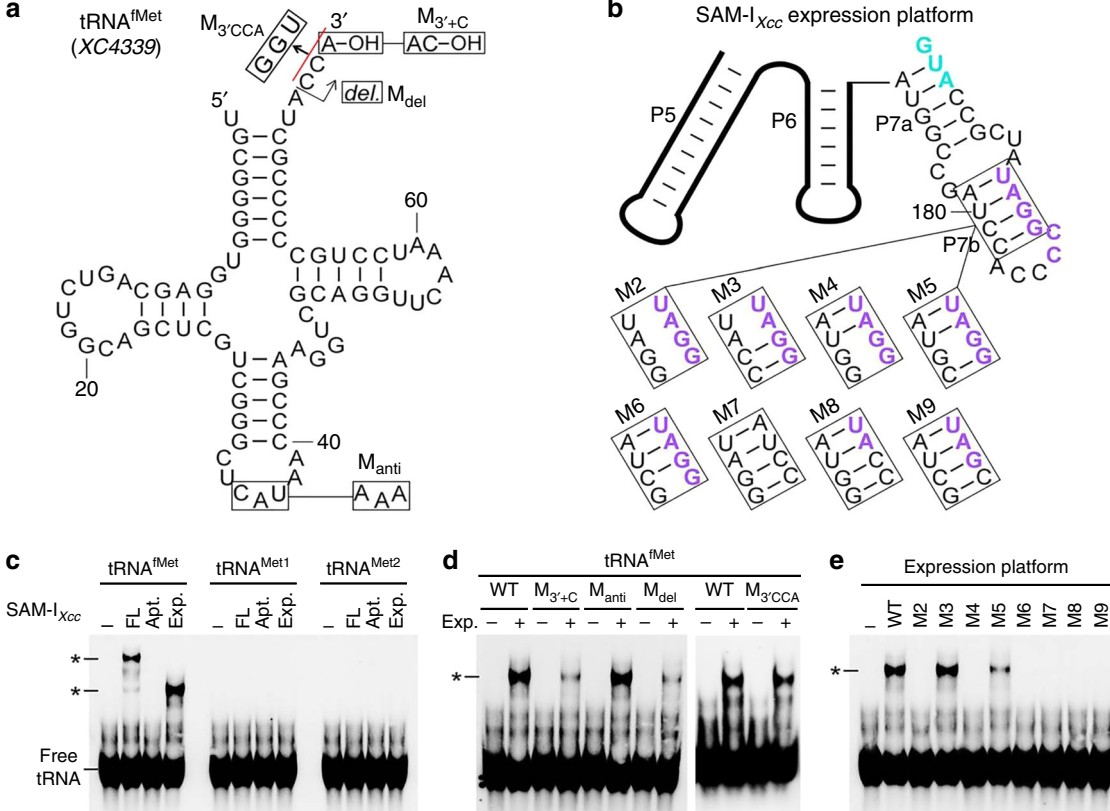

**Fig. 4 The SAM-I$_{Xcc}$ expression platform binds with initiator Met-tRNA (tRNA$^{fMet}$). a** Sequence and secondary structure of *Xcc* tRNA$^{fMet}$. Mutant constructs include the charged tRNA$^{fMet}$ mimic (M$_{3'+C}$), the CAU to AAA mutation in anti-codon (M$_{anti}$), and the deletion of the 3'CCA (M$_{del}$) and the substitution of 3'CCA with GGU (M$_{3'CCA}$). **b** Mutant constructs (M2 to M9) examined in SAM-I$_{Xcc}$ expression platform for the interaction between tRNA$^{fMet}$ and SAM-I$_{Xcc}$. **c** EMSA detection of the specific binding of tRNA$^{fMet}$ to SAM-I$_{Xcc}$ expression platform. The full-length (FL), the aptamer (Apt.), and the expression platform (Exp.) of SAM-I$_{Xcc}$ were examined individually with the uncharged and DIG-labeled Met-tRNAs in *Xcc*, including tRNA$^{fMet}$, tRNA$^{Met1}$, and tRNA$^{Met2}$ (elongator tRNAs). Bands referring to the complex of tRNA and SAM-I$_{Xcc}$ or the expression platform of SAM-I$_{Xcc}$ (*) are pointed out. **d, e** EMSA assay revealing the potential binding site in tRNA$^{fMet}$ and SAM-I$_{Xcc}$ expression platform. Mutant constructs tested are depicted in **a** and **b**. For the assay, same conventions apply as in **c**. Source data are provided as a Source Data file.

expression platform and the charged tRNA$^{fMet}$ mimic (Fig. 4d), implying that SAM-I$_{Xcc}$ is able to discriminate against the charged tRNA$^{fMet}$ and selectively binds with the uncharged tRNA$^{fMet}$. Interestingly, the binding was nearly abolished by the deletion of 3'CCA but not affected while 3'CCA was changed to GGU, indicating that the 3'CCA nucleotide sequence itself is not important but the length of the 3'-end or the overall shape of

tRNA$^{fMet}$ is important for the binding. The mutation in anticodon did not affect its binding with the expression platform (Fig. 4d), suggesting that the anticodon is not important for the binding. These findings reveal that the tRNA recognizion mechanism used by the expression platform of SAM-I$_{Xcc}$ is different from previous characterized T-box[8], in which base pairing interaction between the UGG in the T-box loop and the

tRNA 3′-CCA end, as well as between the codon in the specifier loop and its corresponding anticodon in the tRNA anticodon loop is essential for the recognition[8], but is consistent with the facts that neither T-box loop nor specifier loopis presented in the expression platform of SAM-I$_{Xcc}$ (Supplementary Fig. 14), and that the expression platform can not bind with the elongator Met-tRNAs (tRNA$^{Met1}$and tRNA$^{Met2}$) (Fig. 4c), although their 3′-CCA end and anticodon are identical to that of the initiator Met-tRNA (tRNA$^{fMet}$). Furthermore, deletion of the first nucleotide "U" at the 5′-end, substitution of the 73U74A with 73A74U, mutation in the D -loop or T-loop of the tRNA$^{fMet}$ severely reduced its binding ability with the expression platform (Supplementary Fig. 18), indicating that these regions are important for the binding.

To test whether the anti-SD sequence is the tRNA$^{fMet}$ binding site, a set of assays were carried out to examine the interaction between tRNA$^{fMet}$ and the SAM-I$_{Xcc}$ expression platform carrying a selection of modifications (M2-M6) (Fig. 4b). A complete loss of the binding between the tRNA$^{fMet}$ and the expression platform was observed when multiple (M2, M4) or single nucleotide substitution (M6) were introduced into the anti-SD sequence, while the structural compensatory mutations (M7-9) did not restore the binding ability (Fig. 4e). A single nucleotide substitution (M6) in the expression platform resulted in a complete loss of its binding ability towards the tRNA$^{fMet}$ (Fig. 4e), suggesting that the binding between the tRNA$^{fMet}$ and the SAM-I$_{Xcc}$ expression platform is not due to a gratuitous base-pairing. These data indicate that the anti-SD stem structure is not important for the binding whereas the anti-SD sequence appears to be. In addition, disrupting the anti-SD stem by mutating the SD (M10) did not affect the binding (Supplementary Fig. 19). Collectively, these data support the conclusion that the anti-SD sequence is the binding site of tRNA$^{fMet}$. In addition, a mutation in the anti-AUG sequence (AUGGC, position 172-176), which contains the only 3′CCA complemary sequence UGG in the expression platform, did not affect its binding with tRNA$^{fMet}$ (Supplementary Fig. 18), implying that the anti-AUG sequence is not the binding site of tRNA$^{fMet}$. This data further supports the conclusion that base pairing between the 3′CCA and the UGG is not important for the recongnition. Moreover, the binding of tRNA$^{fMet}$ to SAM-I$_{Xcc}$ was not affected by SAM in vitro (Supplementary Fig. 20), suggesting that the SAM induced structural reorganization of stems P1, P4-6, and P7a of SAM-I$_{Xcc}$ (Fig. 4a) is unlikely related with the binding, which is also consistent with the identification of a critical binding site (C181C182) in the P7b stem.

**tRNA$^{fMet}$-binding destabilizes anti-SD stem and frees the SD**. The binding of tRNA$^{fMet}$ to the position C181C182 of SAM-I$_{Xcc}$ in principle should disrupt the anti-SD stem (P7b) and release the SD. In order to validate this notion, we carried out a selection of RNase H cleavage assays (see Methods)[15]. To achieve this the full-length SAM-I$_{Xcc}$ was hybridized to a short (12 nt) DNA oligo complementary to the SD region (Fig. 3a), followed by treatment with RNase H, which specifically cleaves the RNA:DNA heteroduplex (see Methods). As shown in Fig. 3b, the SD region became more available for the DNA oligo to anneal when the uncharged tRNA$^{fMet}$ was present during the RNA refolding process, regardless of the presence or absence of SAM. This result supports the structural model in Fig. 2a and provides direct evidence that the binding of tRNA$^{fMet}$ to SAM-I$_{Xcc}$ frees the SD sequence. Addition of uncharged tRNA$^{fMet}$ did not affect the cleavage efficiency when a DNA oligo that complementary to another region was used (Supplementary Fig. 12), demonstrating that the effect of tRNA$^{fMet}$ is site-specific.

**tRNA$^{fMet}$-binding derepresses SAM's inhibitory effect in vivo**. Whether the binding of tRNA$^{fMet}$ to SAM-I$_{Xcc}$ influences the expression of the met operon in vivo was further investigated. A recombinant plasmid over-expressing tRNA$^{fMet}$ by the BAD promoter was introduced into Xcc strain to elevate the uncharged tRNA$^{fMet}$ level. Northern blotting analysis revealed that the levels of tRNA$^{Met1}$, tRNA$^{fMet}$, and a mutated tRNA$^{fMet}$ (73U74A → 73A74U, i.e., M$_{7374}$ in Supplementary Fig. 18) in the over-expression strains were over 20-fold higher than that expressed from the chromosomal copy alone (Supplementary Fig. 21), and the majority of the over-expressed tRNA was uncharged (Supplementary Fig. 22). When these strains were cultured in the minimal medium supplemented with 2.5 µM SAM, a concentration that can completely inhibit the expression of the met operon (Supplementary Fig. 23), a significantly increased expression of XC1251, the first downstream gene of SAM-I$_{Xcc}$ in the met operon (Fig. 5a), was confirmed in the tRNA$^{fMet}$ over-expression strain relative to the normal strain by Western blotting (Fig. 5b). No obvious elevation of XC1251 level was observed in the strains over-expressing tRNA$^{Met1}$ and the mutated tRNA$^{fMet}$ (4339 M) that lost the binding ability to SAM-I$_{Xcc}$ (Supplementary Fig. 18). Likewise, only over-expression of tRNA$^{fMet}$ led to an apparent increase of the GUS activity in the gusA translational fusion reporter strain when cultured in the minimal medium supplemented with 250 µM SAM, a concentration that can completely inhibit the GUS activity of the translational reporter strain (Supplementary Fig. 23), compared to the normal expression of tRNA$^{fMet}$, the over-expression of tRNA$^{Met1}$ or over-expression of mutated tRNA$^{fMet}$ (Fig. 5c).

These findings suggest that over-expression of tRNA$^{fMet}$ can partially derepress SAM's inhibitory effect in vivo, consistent with the in vitro observation that the tRNA$^{fMet}$ can independently bind to SAM-I$_{Xcc}$, resulting in the release of the SD for translational regulation. This tRNA$^{fMet}$ controlled regulation via its binding to SAM-I$_{Xcc}$ also finishes the last piece of the puzzle in the regulation process of SAM-I$_{Xcc}$, that is, the uncharged tRNA$^{fMet}$ and SAM act independently on SAM-I$_{Xcc}$ for genetic regulation (Fig. 6).

## Discussion

Riboswitch-mediated gene regulation is one of the most direct and active feedback regulation systems found in bacteria[2,12,37]. The discovery of numerous riboswitch classes has indicated these RNA molecules play critical roles in modulating many bacterial cellular processes including metabolism and virulence[2,12,37]. Riboswitch gene regulation is considered rapid and responsive to changing environmental conditions when compared to protein-mediated regulation[2,12,37]. In this study, we characterized the riboswitch SAM-I$_{Xcc}$ that regulates methionine synthesis in the Gram-negative bacterial pathogen Xcc. We performed several in vitro and in vivo experiments showingthat SAM-I$_{Xcc}$ controls the met operon primarily by modulation of translation in response to cellular levels of SAM. Through a series of biochemical and genetic assays we also demonstrate that the expression platform of SAM-I$_{Xcc}$ is endowed with a dual sensing ability. We specifically demonstrate that besides serving as a classic SAM-I expression platform, which undergo structural change to repress gene expression upon SAM binding to the aptamer, the expression platform of SAM-I$_{Xcc}$ also has the unique ability to sense and bind with uncharged initiator Met-tRNA, allowing the platform itself to modulate translation initiation. As far as we know, the expression platform of SAM-I$_{Xcc}$ is the first riboswitch expression platform validated to have sensing function.

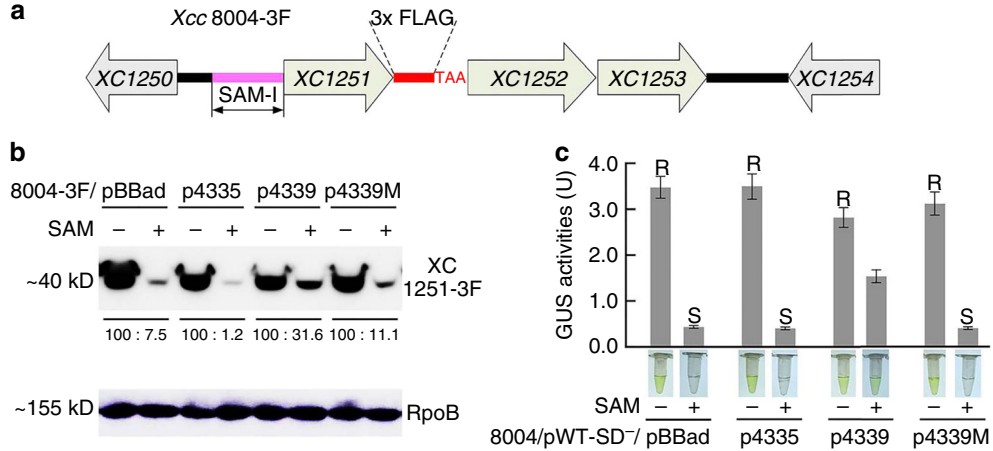

**Fig. 5 Derepression of SAM's inhibitory effect by over-expression of tRNA$^{fMet}$. a** The genetic organization of the *met* operon locus in the recombinant strain *Xcc* 8004-3F. A 3xFLAG-tag and a stop codon (TAA) were inserted between *XC1251* and *XC1252* to monitor the expression of XC1251. **b** Western blot detection of XC1251-3F in strain *Xcc* 8004-3F carrying different plasmids. The culture was grown in MMX alone (SAM−) or MMX supplemented with 2.5 μM SAM (SAM+). pBBad: an expression vector; p4335, p4339, and p4339M: the recombinant pBBad expressing tRNA$^{Met1}$, tRNA$^{fMet}$, and tRNA$^{fMet}$ mutant (73U74A → 73A74U) that lost the binding ability with SAM-I$_{Xcc}$, respectively. RNA polymerase $\beta$ sub-unit (RpoB) protein was used as a control. **c** Plot of GUS activities produced by the reporter strain 8004/pWT-SD− carrying the series of plasmids in **b** in response to SAM. The culture was grown in MMX alone (SAM−) or MMX supplemented with 250 μM SAM (SAM+). Data are presented as mean values ± SD from three biologically independent samples. Different letters above the columns represent the significant difference at $P < 0.05$ by Student's two-tailed *t*-test. Columns marked with the same letter are not significantly different from one another. Representatives of colorimetric samples for each condition were also included in the graph. Source data are provided as a Source Data file.

In general, riboswitches are believed to sense and respond to a single regulatory signal which allows the cell to effectively control gene expression in response to changes in the environment[2,12,37]. Riboswitches are usually unable to carry out sophisticated genetic control because of simple in structure and mode of action. However, a few unusual riboswitches have been reported making sophisticated genetic decisions by increasing structural complexity, such as tandem arrangement of two aptamers or two different classes of riboswitches[1,7,38,39]. Tandem riboswitches can sense two different small molecules (e.g., SAM and AdoCbl)[38] or two identical molecules (e.g., glycine, TPP, and AdoCbl)[1,7,39]. Similarly, some T-box riboswitches have also been identified to occur in tandem and thus can bind two tRNA molecules[40]. Evidence presented here demonstrates that, unlike all known SAM-I riboswitches, SAM-I$_{Xcc}$, a structurally typical SAM-I riboswitch, appears to have the unique ability to respond to two different types of signals, specifically a small molecule (SAM) and an RNA molecule (uncharged tRNA$^{fMet}$) using a complex mechanism.

The data presented indicates that SAM-I$_{Xcc}$ responds to intracellular concentrations of SAM and uncharged tRNA$^{fMet}$. In order to accommodate these specific interactions, we believe that SAM-I$_{Xcc}$ is able to switch between four different states: 'OFF', 'Partial ON 1', 'Partial ON 2', and 'ON' (Fig. 6). It is possible that SAM-I$_{Xcc}$ may have evolved in order to ensure Met supply in situations where the intracellular concentrations of Met and SAM are not collinear. The observed structural and functional flexibility of SAM-I$_{Xcc}$ could provide the bacterial cell with a survival advantage. For example, in the situation when SAM cellular levels are high enough to stabilize an 'OFF' structure of SAM-I$_{Xcc}$, but the Met level is not sufficient to maintain normal protein synthesis, in this case, Met biosynthesis pathway can be partially activated by uncharged tRNA$^{fMet}$ binding directly to the expression platform (Fig. 6). Furthermore, by evaluating intracellular Met status through the measure of two independent signals (SAM and uncharged tRNA$^{fMet}$) allows SAM-I$_{Xcc}$ more thorough sensing and modulating Met metabolism.

In addition to acting as a regulator in response to the binding of SAM to the aptamer, the expression platform of SAM-I$_{Xcc}$ can directly and specifically recognize and bind uncharged tRNA$^{fMet}$. Binding of uncharged tRNA$^{fMet}$ to the expression platform of SAM-I$_{Xcc}$ leads to sequestration of the anti-SD sequence, which frees the SD for translation initiation. The ability of SAM-I$_{Xcc}$ expression platform to sense uncharged tRNA$^{fMet}$ is a previously unrecognized riboswitch trait. The only known tRNA-responsive riboswitches are the T-box family members, which share a highly conserved T-box sequence (binding to the 3′CCA end of tRNA) and a specifier loop (binding to the anticodon of tRNA), and are restricted to Gram-positive bacteria[8,11,22,41]. As far as we know, SAM-I$_{Xcc}$ is the first RNA found to be capable of sensing tRNAs outside of T-box elements. SAM-I$_{Xcc}$ displays no sequence and structure similarity to any known T-box riboswitches and the anticodon of tRNA$^{fMet}$ is not important for SAM-I$_{Xcc}$-tRNA$^{fMet}$ recognition, indicating that the tRNA recognition mechanism used by SAM-I$_{Xcc}$ is different from that used by known T-box riboswitches. For T-box, sequence complementary between tRNA 3′CCA end and the conserved UGG motif in the T-box loop, betweenthe anticodon of tRNA and the cognate codon in the specifier loop of T-box, as well as the overall shape complementarity of both RNA binding partners have been shown to be essential for T-box-tRNA recognition[22,41]. It is possible that both sequence complementarity and overall shape complementarity are essential for SAM-I$_{Xcc}$-tRNA$^{fMet}$ recognition. Notably, unlike the other 19 aa-tRNAs, Met-tRNAs can be further divided into initiator Met-tRNA (tRNA$^{fMet}$) and elongator Met-tRNA (tRNA$^{Met}$)[42,43]. Our data revealed that the expression platform of SAM-I$_{Xcc}$ can selectively bind with tRNA$^{fMet}$ but not tRNA$^{Met}$. Similarly, a Met-RNA-specific T-box in *Staphylococcus aureus* can also selectively bind to tRNA$^{fMet}$ but not tRNA$^{Met}$[20]. How this T-box and SAM-I$_{Xcc}$ distinguish the tRNA$^{fMet}$ from tRNA$^{Met}$ is an attractive topic which remains to be further investigated.

This work has demonstrated that the expression platform of SAM-I$_{Xcc}$ possesses functional traits of both SAM-I and T-box riboswitches described in Gram-positive bacteria. SAM-I$_{Xcc}$ homologs exist and are highly conservedin sequence (more than 90% identity) and secondary structure (Supplementary Fig. 24) in

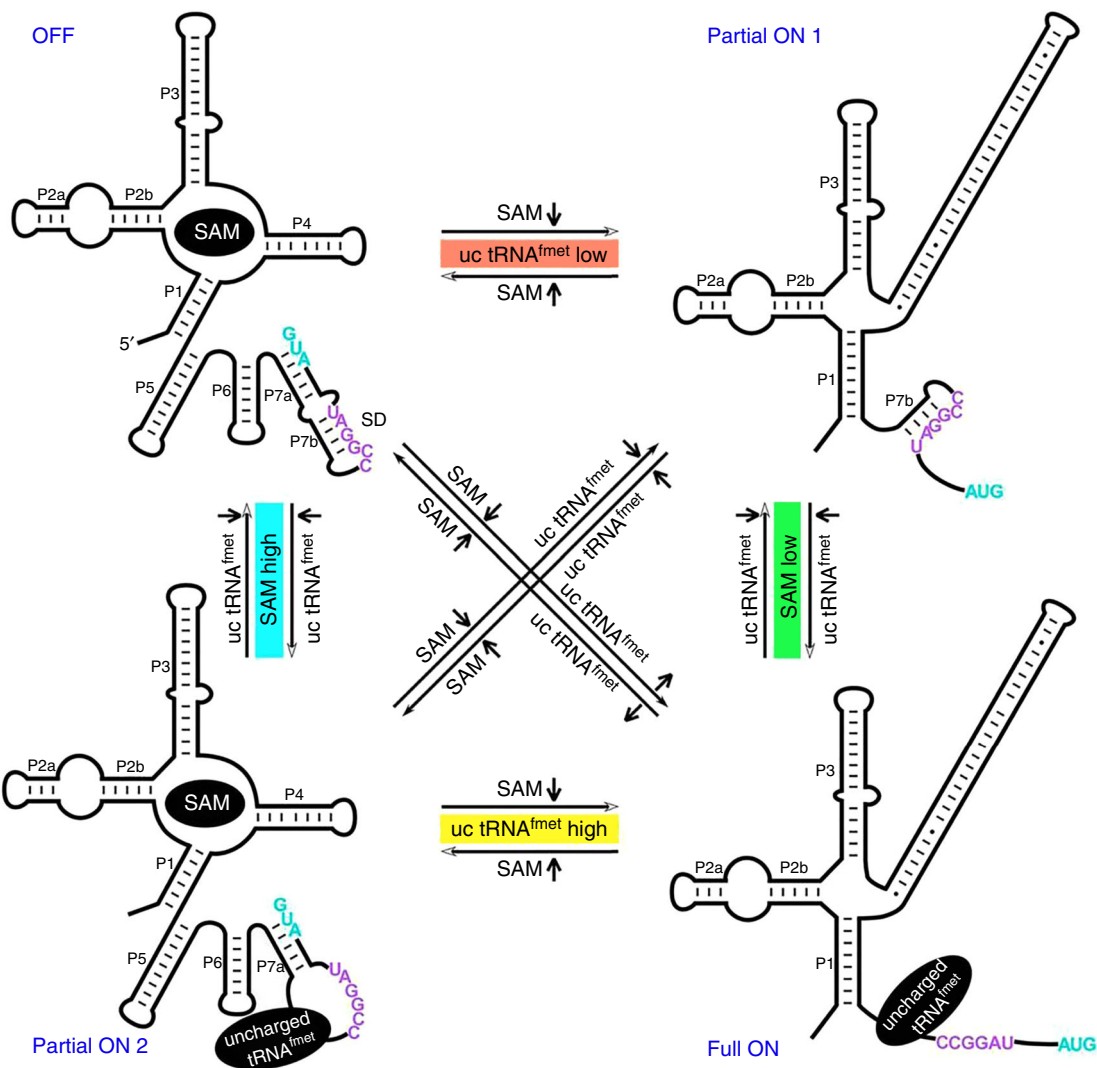

**Fig. 6 A proposed working model for SAM-I$_{Xcc}$ riboswitch.** Upon sensing the fluctuation of SAM and uncharged (uc) tRNA$^{fMet}$, the riboswitch could change its configuration among four states: OFF, Partial ON 1, Partial ON 2, and ON.

the 5′UTR of *metA*, a gene encoding the key Met biosynthesis enzyme homoserine O-acetyltransferase, in nearly all *Xanthomonas* species whose genomes have been sequenced[26], indicating that SAM-I$_{Xcc}$ mechanism may be commonly used by *Xanthomonas* species to control Met biosynthesis. Although there are a few single nucleotide changes in the base-pairing region of the expression platform among different species (Supplementary Fig. 25a), it seems that these changes may not affect its function, since mutating the corresponding nucleotides in SAM-I$_{Xcc}$ did not affect its tRNA$^{fMet}$-binding ability (Supplementary Fig. 25b). Moreover, inactivation of the *met* operon in *Xcc* resulted in Met auxotroph and significantly reduced virulence[26], suggesting that SAM-I$_{Xcc}$ may be a potential target for controlling the diseases caused by *Xanthomonas*.

## Methods

**Bacterial strains and plasmids.** Bacterial strains and plasmids used in this work are listed in Supplementary Table 1. *E. coli* strains were grown routinely in LB medium at 37 °C. *Xcc* strains were grown in the rich medium NYG[44] or the minimal medium MMX[45] at 28 °C. When required, growth media were supplemented with antibiotics at the following final concentrations: rifampicin-50 μg/ml, kanamycin-25 μg/ml, and tetracycline-5 μg/ml.

**Preparation of RNA molecules.** RNA molecules were produced by in vitro transcription using the appropriate DNA templates and T7 RNA Polymerase

(Roche Applied Science, Mannheim, Germany). The corresponding DNA templates were prepared by PCR amplifying the genomic DNA of *Xcc* strain 8004 using specific primers with the promoter sequence (TAATACGACTCACTATAG GG) for T7 RNA polymerase (T7 RNAP) at the 5′-end (see Supplementary Table 2 for primer sequences). RNA molecules with a desired mutation were generated via primer-mediated apporach. In vitro transcription was carried out in 40 mM Tris-HCl (pH 8.0 at 23 °C), 6 mM MgCl$_2$, 10 mM DDT and 2 mM spermidine at 37 °C for 4 h and the resulting transcripts were purified by gel extraction using an RNA gel extraction kit (Shanghai solarbio Bioscience and Technology Co., LTD, Shanghai, China). Digoxigenin (DIG)-labeled RNAs were prepared by using DIG RNA labeling kit (Roche Applied Science, Mannheim, Germany) according to the manufacturer's protocols. To generate $^{32}$P-labeled RNAs, purified RNAs were dephosphorylated using alkaline phosphatase (New England Biolabs) and then 5′ radiolabeled using [γ-32P] ATP (Applied Biosystems, USA) and T4 polynucleotide kinase (New England Biolabs), and the 5′-$^{32}$P-labeled RNAs were isolated by denaturing 6% PAGE and recovered with an RNA gel extraction kit (Shanghai solarbio Bioscience and Technology Co., LTD, Shanghai, China).

**In-line probing reactions.** Due to the high G+C content in the riboswitch, modified in-line probing reactions (higher temperature) were conducted for 36 h at 37 °C, in addition to 22 °C, in mixtures containing 20 mM MgCl$_2$, 100 mM KCl, and 50 mM Tris (pH 8.3 at 23 °C). For each probing reaction, ~50 pM 5′-$^{32}$P-labeled RNA was incubated with added compounds as indicated for each experiment. Partial alkaline digestion of RNA was performed by incubating ~1 nM 5′-$^{32}$P-labeled RNA (typically ~300 kcpm/μl) in a 20-μl mixture containing 50 mM Na$_2$CO$_3$ (pH 9.0 at 23 °C) and 1 mM EDTA at 90 °C for 5–10 min, followed by immediate cooling on ice. RNase T$_1$ cleavage ladder was created by incubating ~1 nM 5′-$^{32}$P-labeled RNA (typically ~300 kcpm/μl) in a 20-μl mixture containing

3 M urea, 25 mM sodium citrate (pH 5.0 at 23 °C) and 2 units of RNase $T_1$ at 55 °C for ~15 min, followed by immediate cooling on ice. RNA cleavage products were separated by denaturing (8 M urea) 10% sequencing polyacrylamide gel (PAGE), which was dried and then visualized using a Typhoon 9000 Phosphor Imager (GE Healthcare). The data on the gel were analyzed using ImageQuant software (Molecular Dynamics).

The $K_D$ value for SAM-I$_{Xcc}$ was determined by performing in-line probing of the full-length RNA construct (aptamer + expression platform) and varying SAM concentration. Bands (R1-4 in Fig. 1d) undergoing SAM-mediated changes in intensity were quantified, and the values were adjusted by subtracting background. The data were further normalized relative to the signal in a band that seems not undergo apparent SAM-mediated changes. The resulting values, termed 'fraction modulated', were scaled from the minimum of 0 to the maximum of 1 and plotted versus the logarithm of the molar concentration of ligand. The data were finally fit to a standard sigmoidal dose-response curve to obtain apparent $K_D$ value.

**gusA reporter constructs**. The experimental strategy and procedure for the construction of SAM-I$_{Xcc}$–gusA fusion reporters were shown in Supplementary Fig. 2. The SD$^+$-gusA and SD$^-$-gusA DNA fragments were amplified by PCR using E. coli k12 genomic DNA as template[46] and the primer pairs SD$^+$-gusA-F/gusA-R and SD$^-$-gusA-F/gusA-R (Supplementary Table 2), respectively. Wild-type SAM-I$_{Xcc}$ DNA fragment from the met operon transcription start site to the translational start site (Supplementary Fig. 1) was amplified by PCR using Xcc 8004 genomic DNA as template and the primer pairs Plac-SAM-I-F/SAM-I-R. Mutated SAM-I$_{Xcc}$ DNA fragments were obtained by the similar PCR amplification procedure with specific primers (Supplementary Table 2). A 17-nt tag was designed in the primers to generate the overlap region at the ends of gusA and SAM-I$_{Xcc}$ DNA fragments to allow them to be further amplified by fusion PCR[47] to construct the SAM-I$_{Xcc}$-SD$^+$-gusA and SAM-I$_{Xcc}$-SD$^-$-gusA DNA fragments, which were cloned into the vector pLAFR6 as an EcoRI-HindIII fragment to generate a transcriptional fusion (SD$^+$) and translational fusion (SD$^-$), respectively.

**GUS reporter assays**. The activity of β-Glucuronidase (GUS) was determined as described by Jefferson et al.[48]. The Xcc reporter strains were grown to mid-log phase (OD$_{600}$ = 0.6–0.7) in 10 ml of the minimal medium MMX with or without supplementation of SAM (to a desired final concentration) at 28 °C with shaking at 200 rpm. Cells were harvested by centrifugation for 10 min at 2.4×g and resuspended in fresh MMX to a cell density of OD$_{600}$ = 1.0. One microliter aliquot was transferred to a 1.5-ml EP tube, and cells were lysed by addition of 40 μl dimethylbenzene and vortexed vigorously for 1 min. Then 125 μl lysate was transferred to a new 1.5-ml EP tube, and 375 μl GUS reaction buffer [consisting of 50 mM sodium phosphate (pH 7.0), 10 mM 2-mercaptoethanol, 0.1% Triton X-100, and 1 mM ρ-nitrophenyl-D$^-$ glucuronide] was added. Reactions occurred at 37 °C for 10 min and were terminated by addition of 200 μl of 2.5 M 2-amino-2-methylpropanediol. ρ-Nitrophenol absorbance was measured at 415 nm using a UV–Vis spectrophotometer (UV-2400PC, Shimadzu, Japan). MMX medium alone was used as a blank. One unit of GUS activity was defined as 1 milligram (mg) of ρ-nitrophenol released from ρ-nitrophenyl β-D-glucuronide per minute per ml of bacterial culture (cell density: OD$_{600}$ = 1.0). The ρ-nitrophenol concentration (mg/ml) was calculated according to the ρ-nitrophenol standard curve showing the absorbance at 415 nm of different concentrations (mg/ml) of ρ-nitrophenol.

**RNA gel mobility shift assays**. DNA templates for generating tRNAs [XC4335 (tRNA$^{Met1}$), XC4339 (tRNA$^{fMet}$), XC4381 (tRNA$^{Met2}$), mutated XC4339 (M$_{3'+C}$, M$_{anti}$, M$_{del}$and M$_{3'CCA}$)] and riboswitch RNAs [SAM-I$_{Xcc}$, the aptamer of SAM-I$_{Xcc}$, the expression platform of SAM-I$_{Xcc}$, the mutated expression platforms (M2-9)] were produced by PCR amplification using Xcc 8004 genomic DNA as template and the corresponding primer pairs listed in Supplementary Table 2. DIG-labeled tRNAs and unlabeled riboswitch RNAs were prepared as described above. To examine the possible interaction between tRNA and SAM-I$_{Xcc}$ riboswitch (or components), the riboswitch RNA (final concentration: 10 μM) was mixed with the DIG-labeled tRNA (final concentration: 1 nM) in 20 μl of 500 mM Tris-HCl (pH 6.8) buffer for 30 min at 30 °C. Following that, the samples were electrophoresed in 10% native polyacrylamide gels at low voltage for certain time to allow the bands to be separated. The gels were then transferred to a positively charged nylon membrane (Roche Applied Science, Mannheim, Germany), and bands were detected by using the DIG-Northern Starter Kit (Roche Applied Science, Mannheim, Germany) and visualized with a ImageQuant LAS 500 imager (GE Healthcare).

**RNase H cleavage experiments**. RNase H cleavage experiments were carried out as described previously[15] with minor modifications. DIG-labeled SAM-I$_{Xcc}$ riboswitch RNAs and unlabeled tRNA$^{fMet}$ were prepared as described above. 10 μl of DIG-labeled riboswitch RNA (1 μM) was incubated at 30 °C for 30 min in 500 mM Tris-HCl (pH 6.8) with or without the addition of unlabeled tRNA$^{fMet}$ (1 μM). Then 10 μl of antisense DNA oligos (100 μM) were added to each sample, and incubated at 30 °C for 10 min. After that, 2 units of RNase H (New England Biolabs) were added to the samples to allow the digestion of DNA/RNA hybrids at 30 °C for 15 min. The products were separated on 10% denaturing (8 M urea) polyacrylamide gel. For the following gel-processing, procedures similar to those in

RNA gel mobility shift assays were used. Signals in the RNA bands were quantified using ImageQuant Software (Molecular Dynamics).

**Construction of Met-tRNA over-expression strains**. The Xcc strains 1251-3 F/p4335, 8004/pWT-SD-/p4335, 1251-3 F/p4339, 8004/pWT-SD-/p4339, 1251-3 F/p4339M, and 8004/pWT-SD-/p4339M (Supplementary Table 1), which over-expressed the elongator Met-tRNA (tRNA$^{Met}$), the initiator Met-tRNA (tRNA$^{fMet}$), or the tRNA$^{fMet}$ mutant (U73A74 → A73U74), were constructed by introducing the plasmid p4335, p4339, or p4339M into the Xcc strains 1251-3 F and 8004 (Supplementary Table 1), respectively. The plasmids p4335, p4339, and p4339M (Supplementary Table 1) were constructed by cloning the tRNA$^{Met}$ gene (XC4335), the tRNA$^{fMet}$ gene (XC4339), and the DNA fragment encoding the tRNA$^{fMet}$ mutant (U73A74 → A73U74), which were obtained by PCR amplification using Xcc 8004 genomic DNA as template and the primer sets OE4335F/OE4335R, OE4339F/OE4339R, and OE4339MF/OE4339MR (Supplementary Table 2), into the expression vector pBBad (Supplementary Table 1), respectively.

**Western blotting**. Xcc cells were cultivated in a desired condition, and then harvested by centrifugation. After removing the supernatant, the cells were resuspended in PBS buffer and lysed by sonication. The cell lysates were boiled and separated using 12% Bis-Tris SDS-PAGE with the Mini-Protean Tetra Electrophoresis System (Bio-Rad) and then electroblotted onto PVDF (0.45 μM, Merck Millipore) membrane with the Trans-Blot blotter (Bio-Rad). The membranes were blocked with 5% Difco TM Skim Milk in TBST and probed with Anti-FLAG M2 monoclonal antibody (1:5000 dilution, Beyotime, China). Signals were detected with ahorseradish peroxidase-linked anti-mouse secondary antibody (Beyotime, China) and SuperSignal West Pico PLUS Chemiluminescent Substrate (Thermo Fisher Scientific). Blots were imaged using the Amersham Imager 600 (GE Healthcare).

**Northern blotting**. Xcc strains were grown in a desired condition and harvested by centrifugation. Total RNA was isolated using the PureLink RNA Mini kit (Thermo Fisher Scientific). Three microgram of total RNA were electrophoresed on a 6% denaturing (8 M urea) polyacrylamide gel and transferred to a positively charged nylon membrane (Roche Applied Science, Mannheim, Germany). After UV-crosslinking, the membrane was hybridized at 68 °C for 8 h with a DIG-labeled RNA probe prepared by using a DIG RNA labeling kit (Roche Applied Science, Mannheim, Germany). Signals were detected using the DIG-Northern Starter Kit (Roche Applied Science, Mannheim, Germany), visualized with the ImageQuant LAS 500 imager (GE Healthcare), and quantified by using the GelQuant.NET software provided by biochemlabsolutions.com.

**Statistics and reproducibility**. The experiments were not randomized and the investigators were not blinded to allocation during experiments and outcome assessment. The Student's two-tailed t-test was performed for comparison of means between two data points. The results presented are from a representative experiment done in triplicate which was repeated at least three times with similar results.

**Reporting summary**. Further information on research design is available in the Nature Research Reporting Summary linked to this article.

## Data availability
The data that support the findings of this study are available from the corresponding authors upon request. The source data underlying Fig. 1a, d, 2c, 3b, 4c, d, e, and 5b,c and Supplementary Figs 3, 4b, 5-9, 11, 12, 16–22, 23a, b and 25b are provided as a Source Data file.

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

## Acknowledgements

We thank professor Zihe Rao (Tsinghua University, Beijing, China) for providing manual DNA sequencing advice for in-line probing analysis, professor Guoliang Xu (Shanghai Institute of Biochemistry and Cell Biology, CAS, Shanghai, China) for offering space and equipment for in-line probing experiments, and professor Sanshu Li (School of Biomedical Science, Huaqiao University, Quanzhou, China) for helpful discussions and constructive suggestions. This work was supported by the National Natural Science Foundation of China (31470237, 91859104, 81861138004, 21673050), the 973 Program of the Ministry of Science and Technology of China (2015CB150600), the Ba Gui Scholar Program of Guangxi Zhuang Autonomous Region of China (2014A002), and the Outstanding Clinical Discipline Project of Shanghai Pudong (PWYgy2018-08).

## Author contributions

J.-L.T., H.G., and D-J.T. designed the experiments. H.G. and D-J.T. constructed the structure model of SAM-I$_{Xcc}$. D.W. and W.-Y.Z. constructed some -*gusA* reporters and performed some GUS assay experiments. H.G., X.D., Q.S., J.H., and Y.Z. performed in-line probing analysis and interpreted the data. Y.-P.H. performed Western blotting analysis. J.-Y.L. performed Northern blotting analysis. Z.M., Y.-M.C., and Y.-W.L. contributed to -*gusA* reporter construction and GUS assays. D.-J.T. and J.-L.Z. performed all other experiments, including RNA preparations, generation of SAM-I$_{Xcc}$ and tRNA$^{fMet}$ mutations, electrophoretic mobility shift assays, -*gusA* reporter constructions and GUS assays. S.-Q.A. interpreted the data and revised the manuscript. J.-L.T., H.G., and D-J.T. wrote the manuscript.

## Competing interests

The authors declare no competing interests.
