## [Peer Review File · Nature Communications]

Reviewers' comments:

Reviewer #1 (Remarks to the Author):

In this manuscript, the authors present a novel hypothesis that a particular leader region acts as a classical SAM riboswitch while also associating with tRNA-fMet through an intermolecular interaction that counteracts the riboswitch regulatory mechanism. The manuscript needs to be heavily edited for clarity. While the experiments are interesting, they are not sufficiently convincing for the claims that the authors make in this manuscript. While the data showing that the SAM riboswitch inhibits translation are convincing, the argument that tRNA-fMet also associates with the riboswitch expression platform in a functionally meaningful way is not convincing. More experimentation is required to support these claims.

Comments:

1. Line 10, page 3. The use of the phrase “response regulator element” will be considered confusing by some or many readers as it is typically associated with the response regulator proteins that participate in two component regulatory systems.
2. Page 4, line 12. Change “exist only” to “exist primarily”. While T box systems have been described as being primarily used by gram-positive bacteria, a few prior publications have identified gram-negative examples (Vitreschak et al., 2008). This is incorrectly stated in the discussion section as well.
3. page 4, line 15-16, “recent work examining the regulation of met biosynthesis in the phytopathogen....ref 27 provided the....” This wording seems to imply that the ref 27 specifically investigates methionine biosynthesis in this organism but it seems to be a very general reference on that organism, having nothing specific to do with methionine.
4. Page 5, line 17-18. What is meant in the statement that “SAM-IXcc exhibits no sequence similarity to any known SAM-I riboswitches or any other characterized riboswitches”? While aptamer domains can be meaningfully aligned and assessed for sequence similarity, this isn't really possible for “expression platforms”, which is a term that simply refers to all of the sequence region downstream of the aptamer. That being said, one can assess expression platform sequences for characteristic/diagnostic regulatory elements, such as intrinsic terminators or helical elements near the ribosome binding site. Is this what the authors mean when they say that the SAM-IXcc does not show sequence similarity to known riboswitches?
5. page 4, lines 15-19. Is it true that this is the first example of a 5' leader that controls methionine in a gram-negative? Ribosome-mediated attenuation regulatory mechanisms are common in gram-negative bacteria. Are none of them involved in methionine? I recommend looking more closely at the literature, including Vitreschak et al., 2006.
6. The very small change in the transcriptional reporter upon addition of SAM (Figure 1) seems to argue against transcriptional regulation. For example, the authors cannot rule out other indirect effects, such as the possibility that a minor influence on mRNA stability might be caused by binding of SAM to the aptamer.
7. Page 7, line 5. Choose different wording instead of “fall back”
8. Page 9. The authors seem to be suggesting that the putative hairpins downstream of the aptamer are

acting as an alternate transcription attenuation element. However, alternative explanations are possible, including effects on RNase cleavage, transcriptional pausing, Rho termination. Additionally, alterations in translation can lead to moderate changes in mRNA abundance; high translation can extend the half-life of transcripts relative to the same transcript undergoing lower translation rates. So, while the authors may be correct in concluding that a novel transcription attenuation mechanism is responsible for the moderate changes in gene expression, it is also possible, and I would argue much more likely, that the minor changes in transcript abundance result from indirect causes instead.

9. Page 9, line 24-25: The authors state that constructs containing all three putative hairpins demonstrated ~50% reduction in a transcriptional fusion to the reporter gene. However, Figure 1 shows a ~24% reduction and the secondary structures in Figures 1 and 2 appear to be the same, so I am led to believe that both experiments used constructs that contained all three hairpins. Why are the values different?

10. I am also confused about the authors' interpretations of Figure 2 (page 10). The authors show that the riboswitch mediates strong repression in the presence of SAM. To go with this, they also present a reasonable secondary structure model, which predicts small base pairing segments that pair with the SD and AUG. Mutational disruption alleviates most of the repression when either half of the anti-translation helix is disrupted. When both mutations are combined, full levels of expression are observed. So, these data seem to confirm their regulatory model, yet the authors conclude the opposite. Their secondary structure model can only ever be a guess, because there are many secondary structure models that would be predicted to occur and the authors have not supported the structure model with either biochemical probing (beyond in-line probing, which is consistent with their model but does not present enough data in this region of the sequence to be conclusive) or comparative sequence alignments. That's not a criticism, just a statement. Indeed, I feel their mutational data largely support their secondary structure prediction. The authors appear to be expecting too much of a linear relationship between small helical elements and gene expression, and are ignoring all of the many other factors that will participate in the overall regulatory mechanism. Therefore, the authors must reconsider their interpretation of their own data and fully rework this section.

11. page 10, line 18-19: I was also specifically confused by the authors' interpretation of the M1+M2 mutant, where the authors state that the mutant "showed similar GUS reporter activity...indicating that translation initiation was abolished." Yet, translation levels are literally at full levels. The authors appear to be wrong in this statement.

12. The authors' hypothesis that tRNA fMet associates directly with the leader region RNA is bold. It should therefore be supported by solid proof. The authors present interesting evidence of intermolecular interactions between the leader RNA and the tRNA via EMSA. However, there is an inherent risk in potentially overinterpreting these data. When individual RNA molecules are mixed together at potentially high concentrations (~10 micromolar), it is not unreasonable to expect gratuitous interactions if the RNA molecules share even short segments of base-pairing potential. Their data suggests that this pairing interaction involves nucleotides near the 3' terminus, and does not require the anticodon. The former is potentially interesting; the latter is confusing. How would the leader region avoid associating with the other cellular tRNAs if specificity determinants are not required? And, by extension, the difference in EMSA shifts for initiator and elongation tRNAs may simply result from a loss in the gratuitous base-pairing interaction due to sequence differences. The authors need to, at

minimum, include experiments that involve only single-stranded segments of the tRNA. It is possible that all is needed to recapitulate the EMSA shift they've observed is to include a single-stranded sequence stretch from the 3' region of the tRNA. And if that short single-stranded RNA sequence was equally capable of forming base-pairing interactions with the leader RNA, would the authors consider that meaningful, or would they then consider the possibility that it results from a gratuitous interaction? By extension, the RNase H assays also does not rule out a gratuitous base pairing interaction between tRNA and leader RNA.

13. Page 13, line 15: correct spelling of "colectelly". Perhaps the authors meant "collectively". On a related note, the manuscript needs to be heavily edited for grammar and language throughout, with particular attention to the methods section.

14. Page 16, lines 9-10: I don't think the authors actually mean to use "minimal inhibitory concentration (MIC)", which is a term that means something entirely different than how the authors are using it.

15. In contrast to the dogmatic language used on page 17, I'm not convinced that the experiment shown in Figure 5 alone gives enough proof to state that "these findings clearly indicate that over-expression of tRNA-fMet can derepress SAM's inhibitory effect in vivo...". The authors cannot rule out indirect effects from tRNA overexpression, for example.

Reviewer #2 (Remarks to the Author):

Review for :

Novel variant riboswitch with a sensory and regulatory dual functioning expression platform

The manuscript by Tang et al. presents a SAM-I riboswitch from *Xanthomonas campestris* which appears to have an expression platform that also acts as a sensor for uncharged tRNA^{fMet}. This is a very exciting discovery, which unfortunately the authors do not present too well in my opinion by focusing too much on the so-called "variant riboswitch", rather than on the new tRNA sensing ability of the expression platform, the latter may be a very important contribution for amino acid biosynthesis regulation and may lead to several groups around the world looking for such motifs in other organisms. This would be one of my major comments, which would require quite a bit of re-writing. Notably, "variant riboswitch" normally is used for an aptamer domain that diverges from the consensus. I would instead put more emphasis on what is basically a new tRNA-sensing ncRNA and the fact that it is combined in a dual sensing which requires low SAM concentration and high uncharged tRNA^{fMet} concentration for optimal expression.

My second major comment is that a few experiments would be more than welcome to help further validate and characterize the tRNA sensing switch. More details are provided below, together with many additional comments, in order of appearance in the main text, and then in the supplementary material. That being said, in general the work is well performed with high quality results supporting most of the author's claims. In other words, following some re-writing and a few experiments, I recommend this paper for publication.

Here is a non-exhaustive list of corrections and additional comments (including some important concerns that absolutely need to be looked into):

P2 Line 14: bind with uncharged -> bind uncharged

P3 line 9: changes in response to the changes in the aptamer. - > changes in response to the stabilization of the aptamer structure.

P3 line 18: the first sentence needs at least one reference.

P4 line 5: was controlled -> was discovered to be controlled (or "...is in large part controlled...", depending on the sense the authors want to convey)

P4 line5-6: this sentence sounds incomplete, rephrase

P4 line 7: as a co-activator -> as an inducer

P4 lines 9-11: the authors should add the following reference:

Semen A. Leyn, Inna A. Suvorova, Tatiana D. Kholina, Sofia S. Sherstneva, Pavel S. Novichkov, Mikhail S. Gelfand, Dmitry A. Rodionov (2014) Comparative Genomics of Transcriptional Regulation of Methionine Metabolism in Proteobacteria. PLoSONE.

In fact, the authors should read this paper thoroughly and amend their manuscript in many places accordingly. In particular, they may realize that the intergenic region they are studying includes a "SamR" motif (as described in Leyn et al) specific to *Xanthomonas* species (and perhaps also a MetR motif). This may help explain some results they discuss at the end of the manuscript, but overall does not affect significantly the conclusions of their work.

As soon as the gene XC1251 is mentioned, its function should be mentioned in the main text (homoserine O-acetyltransferase).

P4 line 13: there are actually many SAM riboswitches in Gram negative bacteria, including SAM-II, SAM-V and SAM-SAH (although these only regulate metK, Weinberg et al. 2010, Genome Biol). This information can be found in Rfam, including the SAM riboswitch described in this paper (which is a reason why calling it "a novel variant" seems inappropriate). Because of this, I suggest the authors double-check to see if indeed no other SAM riboswitches were functionally characterized in Gram negatives before (although it would be simpler to delete this statement, since it is not the most important contribution of this work in any case).

P4 line 16: provided the first example of a -> provided evidence of

P4 line 18: of this amino acid -> of Met

P4 line 19: importance -> concern

P4 line 20: It is not clear what “Despite this observation” refers to given the text structure (eventhough we can easily guess). The sentence could simply start with “The mechanism...”

P4 line21: demonstrating this -> demonstrating that this

P4 line 21-22: As stated before, this SAM riboswitch (the aptamer domain) was already predicted and could be found in Rfam’s annotations, so saying that it has “structural similarities to the SAM-I riboswitch class” is an understatement, it was already predicted through homology searches as a SAM-I.

P4 line 23: “Our approach...”, add some details, just a few words to make the sentence and statement more precise.

P5 line 4: platform functions -> platform also functions

P5, end of introduction: here would be one of the good places to better emphasize the findings regarding the new tRNA-sensing RNA.

P5 line 13: the amino acid -> Met

P5 line 17-18: authors state that no sequence similarity to any known expression platform was found. How have the authors verified this? Except for a comparison with *Bacillus subtilis*, there is no sequence alignment in this manuscript, I will come back to this later.

P5 line 19-22: even if one characterized example of a given riboswitch class has a transcription terminator-based expression platform, it is not assumed that all other riboswitches of that same class have the same type of expression platforms, the latter tend to vary more than aptamer domains. Authors may wish to refer to ref 24 in that regard. Also, transcription termination can also occur through a Rho-dependant pathway, more common in gram-negatives (the “consensus” motifs recognized by Rho are C-rich, which would fit with the loops of P5, P6 and P7 in the model provided in Fig1).

P6 line 2: known bacteria -> known that bacteria

P6 line 3: the reference 29 relates to yeast, which forces us to reconsider the statement of that sentence. Another reference should be used.

P6 line 5-6 and 8: the exact GUS expression numbers are provided in the legend, it is redundant in the text (but % variation can be kept).

P6 line 10: the alterative amino acid glycine at a concentration of 0 uM or 300 uM ->

(5 corrections)

an alternative amino acid, glycine, at a concentration of 300 uM

p7 lines 8-9: again, this prediction was already found in Rfam.

P9 line 8: D – (d)

P9 line 12: There is no description of the modeling (neither in the results, nor in the methods, not even in supplementary material). Some brief description should be provided, even if it was performed manually by some of the authors.

P9 line 17: constructs are introduced -> constructs were introduced

P9 line 18: -> without SAM supplementation.

P9 line 23 and 26-27: the way the text is written, it implies that SAM downregulates transcription, but this is not the case as shown by the authors in suppl fig 9 (the +/- SAM pairs always have the same letter, meaning there is no significant difference between each). Even if some mutants do seem to affect transcription. The text should be changed to ensure this is clear. Also, the notation of the GUS constructs with regards to translational vs transcriptional fusions are not always clear (they should have the SD+ or SD- label, as in the text).

P10 line 4: add (Fig. 1) at end of sentence.

P10 line 4: -> The potential mechanism by which SAM-IXcc inhibits...

P10 line 4-5: -> that inhibit translation initiation. The in-line probing experiments revealed...

P10 line 18: and release -> and releases

P10 line 18: I assume that instead of “wild-type”, the authors mean “with or without SAM”, also, the following text would be better as “... indicating SAM-mediated modulation of translational initiation was abolished”

P10 line 19-20: mutant that carries changes that expose the SD -> mutant that that exposes the SD

P10 line 22: -> construct that exposes the start codon AUG...

P11 line 1: mediation -> modulation (or repression?)

It is not clear what the authors mean here, in part because according to the models in figure 2, the SD is still sequestered, even if

P11, figure 2a: add the orange highlight (for anti-AUG) to the “SAM –” version of the structure.

P12 line 4-5: change sentence to: “The homoserine O- acetyltransferase is connected to Xcc Met metabolism pathway and, by extension, to uncharged/charged Met-tRNA.

P12 line 6-7: was too interacted with -> was to interact with

P12 line 7: method -> mechanism

P12 line 8: have several features -> have features

P12 line 10: can interact Met-tRNA -> can interact with Met-tRNA

P12 line 15: was -> were

P12 line 17: -> Additionally, the aptamer domain and the expression...

P13 line 15: Colectelly -> Collectively

P13 line 16: While several mutants were analyzed for the anti-SD, none were analyzed for tRNA binding in the anti-AUG (P7a). This is surprising, given that such a mutant has been tested (Fig. 2 for SAM-mediated regulation of GUS activity). It would also seem particularly interesting to do such a mutant given that this short sequence encompasses a “UGG” sequence, which would be an ideal target for the tRNA-CCA. This would also allow to perform a complementation with a corresponding mutant tRNA. Such an experiment would provide additional support to the tRNA-sensing RNA region the authors uncovered, as well as provide a more detailed model of how this interaction can occur. Clearly, a lot of work would be required to understand in details how this RNA structure distinguishes tRNA-fMet from other tRNAs, but this presumably simple experiment would still be a great help to establish a “basic model” that may serve as the foundation of further research.

P14 line 8: -> methionine-tRNAs

P14 line 10: were -> are

P14 line 11: were -> are

P16 line 10: “MICs” usually express growth inhibition, to prevent confusion, authors are invited to rephrase

P16 line 17: when saying “the MIC condition of SAM”, it is not clear what the authors refer to, two concentrations have been used. Maybe what they mean is “at the corresponding MIC condition”. Note that the use of MICs here is a bit unusual. In the past, similar riboswitch studies have been

performed in conditions where there is an excess of the ligand, why the authors have not used such an excess? Also, if they have, it would be interesting to know what were the results, perhaps this was to have a more “borderline” expression level that could more easily “switch” according to tRNA expression? If it is the case, it would be interesting to know and it would actually add to the interest of the story by providing further evidence of the complex and subtle interplay between different sensing mechanisms.

P17 line 16: -> can partially derepress

P17 line 20: mediation -> regulation

P19 line 2: as mentioned, the claim of “we identified and characterized a variant riboswitch” should be amended, both because of the “identified” (the SAM riboswitch was already predicted by Rfam) and the term “variant riboswitch”. In contrast, the characterization is obviously very significant.

Note that from line 4 to 13 there is some redundancy and some sentences could consequently be simplified.

P19 line 4: We detail -> We performed

P19 line 6: -> we also demonstrate

P19 line 7: delete “whose”

P19 line 7: with previously uncharacterized dual ability. -> with a dual sensing ability.

P19 line 11-13: it would be good to look if what the authors found is the first riboswitch expression platform which also has a sensing function, if so, a statement could be made regarding that (and even if there are other examples, it will likely be one among very few such regulatory regions)

P19 line 16-19, sentences are not clear and require references

P19 line 20-22: what the authors describe as “variants” are rather different combinations of riboswitches in the same UTR, which is also what their manuscript describes, in addition to describing a new type of tRNA-sensor.

P19 line 23: This -> The

P19 line 24: change phrasing to avoid the term “variant”

P20 line 15: “more accurate”, perhaps “more thorough sensing” would be more appropriate?

P20 line 25: t-box elements have been predicted in proteobacteria before, so this is not the first

example of a tRNA-sensing element in Gram negative bacteria. However, it is, to my knowledge, the only RNA capable of sensing tRNAs outside of t-box elements, which is a very exciting finding (and the fact that it is found in the expression platform of a SAM ribowitch is a bonus to make it even more interesting).

P21 line 4: -> For T-box, sequence complementarity...

P21 line 6: box loop; between the anticodon...

P21 line 6: delete existed

P21 line 7, delete "and fitness"

P21 line 7, partners are essential -> partners have been shown to be essential

P21, Line 9: complementarity and overall shape fitness are -> and overall shape complementarity are

P21 line 10: unlike other -> unlike the other

P21 line 13-15 rephrase sentence

P21 line 18: the term "widespread" does not seem appropriate, given that it is apparently only found in Xanthomonas. It would also be interesting to know if it is found in all Xanthomonas species and only upstream of this gene, or also genes encoding other activities (it is of interest that Xcc encodes two homoserine O-acetyltransferases).

P21 line 21: More importantly -> Moreover

With regards to conservation, this paper lacks a lot of details. An alignment should at least be presented in supplementary material, especially given that authors claim the motif is conserved. I have made such an alignment following a few BLAST searches, here is a summarized alignment with non-redundant sequences:

```
Query 1 ACGCGAGCTCCCGCGAAG.CTCGATGGCCGATCCACC..CCGGATATCGCCATG 51
CP000050.1 1531183 ACGCGAGCTCCCGCGAAG.CTCGATGGCCGATCCACC..CCGGATATCGCCATG 1531233
CP016830.1 1274832 ACGCGAGCTCCCGCGAAG.CTCGATGGCCGATCCACC..CTGGACATCGCCATG 1274882
CP018728.1 4131786 ACGCGAGCTCC-GCGAAG.CTCGATGGCCGATCCACC..CCGGATACCGCCATG 4131737
CP019515.1 3062879 ACGCGAGCTCC-GCGAAG.CTCGATGGCCGATCCACC..CTGGATACCGCCATG 3062830
CP020971.1 4621205 ACGCGAGCTCC-GCGAAG.CTCGATGGCCGATCCACC..CTGGATACCGCCATG 4621254
CP034653.1 1907635 ACGCGAGCTCC-GCGAAG.CTCGATGGCCGATCCACC..CTGGATACCGCCATG 1907684
CP020987.1 934400 ACGCGAGCTCC-GCGAAG.CTCGATGGCCGATCCACC..CTGGATACCGCCATG 934351
CP013004.1 1576411 ACGCGAGCTCC-GCGAAG.CTCGATGGCCGATCCACC..CTGGATACCGCCATG 1576460
CP022270.1 3461809 ACGCGAGCCCC-GCGAAG.CTCGATGGCCGATCCACC..CTGGATACCGCCATG 3461760
```

CP013679.1 1768456 ACGCGAGCTCC-GCGAAG.CTCGATGGCCGATCCACC..CTGGATACCGCCATG 1768407
CP019797.1 2553242 CCCGCGAAG.CTCGATGGCTGCTCCTTA..CCGGATATCGCCATG 2553201
CP033586.1 1002918 ACGCGAGC-CCCGCGAAG.CTCGATGGCCGATTCCCC..CCGGATCCCGCCATG 1002970
LS483406.1 3498410 ACGCGAGC-CCCGCGAAGgCTCGATGGCCGATTCCCCaaCCGGATCCCGCCATG 3498358
..... “* ****” /”””***** “ * * * * * *****

Colors at the top highlight the base pairing regions P6 (green), P7a (yellow) and P7b (turquoise). Positions that vary in the alignment are also colored (red=changes that disrupt base pairs; grey= “silent mutations, or gap”; blue=mutations that still permit base pairing; pink=mutations that extend base pairs). As seen from this, there is very little support for the proposed structure model with regards to covariation. However, given that the sequence is indeed very conserved (and many other sequences from other strains and species identical to these ones), this is not so surprising, but does require mutational analysis to confirm the structure and important elements of it, which the authors have done at least in part.

P22 line 15: MgCl₂ (subscript)

P22 line 25: no data were presented for in-line probing at room temperature, this might be interesting to have to compare patterns, especially given that Xcc does not normally grow at 37C as would a human pathogen. Also, rather than GC content, the reason why it was difficult to get modulation at room temperature might be that the full expression platform is tested together with the aptamer. Often aptamers are tested alone, when tested with their expression platform it is not uncommon to lose modulation as the binding of ligand is often co-transcriptional, meaning that the ligand has the possibility of binding even before the expression platform is fully transcribed.

P23 gus reporter constructs: more details would be welcome in suppl Fig. 2 (the sequence from promoter up to the first few codons of gus; for both the SD+ and SD- constructs).

P24 line 10: lysised -> lysed

P24 line 18: milligrams -> milligram

P25: I have never seen “none-labeled”, unlabeled

P26 line 15: in the PBS -> in PBS

P26 line 19: was -> were

P26 line 21: the -> a

P27 line 3: the -> a

P27 line 18: author -> authors

Supplementary material:

Most importantly, some info with regards to sequence conservation should be provided (and at least some mention to what extent it supports the structure model proposed).

Figures of the two other tRNA-Met tested should be presented

S3 line 5: aend -> end

S6: it is not possible to determine a K_d below ~ 1 nM by in-line probing because at lower concentrations, there is more RNA than ligand molecules, which would falsely give the impression that the RNA is not modulating (as compared to absence of ligand). Therefore, the concentration of 10 pM, and even 100 pM are not very useful.

P13-14 this is strangely organized and could not find where it fits in the main text, reorganize

P15 line 5: gnome -> genome

P15 line 8: red -> purple

P15, the reference Leyn et al 2014 should probably be used to improve this figure. (or at least be cited)

Fig S11: part (A) of this figure is taken integrally from Gutiérrez-Preciado et al. (2009) Biochemical Features and Functional Implications of the RNA-Based T-Box Regulatory Mechanism. MICROBIOLOGY AND MOLECULAR BIOLOGY REVIEWS.

Unless authors had consent from the journal and authors, this should be taken out of their manuscript (or they should ask for the permission and, if they get it, indicate that the figure was taken from that article, and not merely cite it).

P22: other factors could explain the difference in regulation, for instance, the plasmid is multicopy. Also, authors might want to discuss the presence of a conserved "Xanthomonadales" box in the promoter region of XC1251, named SamR by (Leyn et al 2014).

Rebuttal letter for NCOMMS-19-17444

Point by point response to reviewers

Reviewer #1 (Remarks to the Author):

In this manuscript, the authors present a novel hypothesis that a particular leader region acts as a classical SAM riboswitch while also associating with tRNA-fMet through an intermolecular interaction that counteracts the riboswitch regulatory mechanism. The manuscript needs to be heavily edited for clarity. While the experiments are interesting, they are not sufficiently convincing for the claims that the authors make in this manuscript. While the data showing that the SAM riboswitch inhibits translation are convincing, the argument that tRNA-fMet also associates with the riboswitch expression platform in a functionally meaningful way is not convincing. More experimentation is required to support these claims.

Response: According to your and the other reviewer's comments, the manuscript has been revised. Several additional experiments suggested by you and the other reviewer have been performed and the results have been added into the revised manuscript. Please see details below. We greatly appreciate your helpful suggestions which have improved the manuscript a lot.

Comments:

1. Line 10, page 3. The use of the phrase "response regulator element" will be considered confusing by some or many readers as it is typically associated with the response regulator proteins that participate in two component regulatory systems.

Response: We have revised the phrase "response regulator element" throughout the article (page 3, lines 8-10 in the revised manuscript).

2. Page 4, line 12. Change "exist only" to "exist primarily". While T box systems have been described as being primarily used by gram-positive bacteria, a few prior publications have identified gram-negative examples (Vitreschak et al., 2008). This is incorrectly stated in the discussion section as well.

Response: We have revised the text to amend for this oversight. The words "exist only" has been changed to "exist primarily" and reference to the manuscript Vitreschak et al. (2008) has been included (Ref. 25) (page 4, lines 12-13 in the revised manuscript). In addition, the inappropriate statement in discussion has also been revised.

3. page 4, line 15-16, "recent work examining the regulation of met biosynthesis in the phytopathogen....ref 27 provided the...." This wording seems to imply that the ref 27 specifically investigates methionine biosynthesis in this organism but it seems to be a very general reference on that organism, having nothing specific to do with methionine.

Response: Reference 27 is a general introduction to the pathogen *Xanthomonas campestris*. We agree that including it here may lead to confusion. We have amended the text and removed the reference (page 4, line 15 in the current version).

4. Page 5, line 17-18. What is meant in the statement that “SAM-IXcc exhibits no sequence similarity to any known SAM-I riboswitches or any other characterized riboswitches”? While aptamer domains can be meaningfully aligned and assessed for sequence similarity, this isn't really possible for “expression platforms”, which is a term that simply refers to all of the sequence region downstream of the aptamer. That being said, one can assess expression platform sequences for characteristic/diagnostic regulatory elements, such as intrinsic terminators or helical elements near the ribosome binding site. Is this what the authors mean when they say that the SAM-IXcc does not show sequence similarity to known riboswitches?

Response: We apologize for the confusion here. We have removed the sentence. “However, the putative expression platform encoded by SAM-I_{Xcc} exhibits no sequence similarity to any known SAM-I riboswitches or any other characterized riboswitches” has been removed from the current version of the manuscript (page 5).

5. page 4, lines 15-19 . Is it true that this is the first example of a 5' leader that controls methionine in a gram-negative? Ribosome-mediated attenuation regulatory mechanisms are common in gram-negative bacteria. Are none of them involved in methionine? I recommend looking more closely at the literature, including Vitreschak et al., 2006.

Response: We looked very closely at related literature and are aware that RNA regulatory elements in upstream regions of *metXW* and *metZ* genes in some Gram-negative bacteria have been previously predicted (Vitreschak et al., 2004; Leyn et al., 2014). However, none of these have been functionally characterized to date. To be more clear about this we have changed the statement “Little information is available regarding the potential riboswitch regulation of Met biosynthesis genes in Gram-negative bacteria” (page 4, lines 10-11 in the previous version) to “Although potential riboswitches involved in the regulation of Met biosynthesis genes have been proposed in Gram-negative bacteria, none of them has been functionally characterized” (page 4, lines 10-12 in the current version). This is more correct and avoids any confusion. In addition, the statement “provided the first example of a Gram-negative bacterium utilizing a 5'UTR region to control the expression of the genes involved in the generation of this amino acid” (page 4, lines 16-18 in the previous version) has been changed to “provided functional evidence of a Gram-negative bacterium utilizing a 5'UTR region to control the expression of the genes involved in the generation of Met” (page 4, lines 15-17 in the current version).

6. The very small change in the transcriptional reporter upon addition of SAM (Figure 1) seems to argue against transcriptional regulation. For example, the authors cannot rule out other indirect effects, such as the possibility that a minor influence on mRNA stability might be caused by binding of SAM to the aptamer.

Response: Based on the data shown in Fig. 1, we cannot rule out the possibility that the SAM-responsive GUS activity repression is caused by an effect on mRNA stability or other indirect effects. Thus, the conclusion that SAM-I_{Xcc} also controls gene expression directly at the transcriptional level is inappropriate. In the revised manuscript, the previous statements “The data indicate that SAM-I_{Xcc} is specifically responsive to the cellular levels of SAM” and “In addition, SAM-I_{Xcc} appears to control gene expression at both the transcriptional and translational levels and the influence at the translational level appears dominant under the conditions tested”

(page 6, lines 10-14 in the previous version) has been changed to “The data indicate that SAM-I_{Xcc} is specifically responsive to the cellular levels of SAM and controls gene expression primarily at the translational level. Additionally, the GUS activity of the transcriptional fusion reporter strain (8004/pWT-SD⁺) shows a small but statistically significant reduction upon addition of SAM, suggesting that SAM-I_{Xcc} may also modulate gene expression weakly at the transcriptional level. However, we cannot rule out that this reduction of GUS activity may be caused by an influence on mRNA stability induced by the binding of SAM to the aptamer” (page 6, lines 8-14 in the current version)

7. Page 7, line 5. Choose different wording instead of “fall back”

Response: We have amended this text (page 7, line 5).

8. Page 9. The authors seem to be suggesting that the putative hairpins downstream of the aptamer are acting as an alternate transcription attenuation element. However, alternative explanations are possible, including effects on RNase cleavage, transcriptional pausing, Rho termination. Additionally, alterations in translation can lead to moderate changes in mRNA abundance; high translation can extend the half-life of transcripts relative to the same transcript undergoing lower translation rates. So, while the authors may be correct in concluding that a novel transcription attenuation mechanism is responsible for the moderate changes in gene expression, it is also possible, and I would argue much more likely, that the minor changes in transcript abundance result from indirect causes instead.

Response: As mentioned in our response to comment #6, the conclusion that SAM-I_{Xcc} also controls gene expression at the transcriptional level has been modified and caveats put in place. We agree that the current data presented cannot rule out the possibility that the SAM-responsive GUS activity repression could be caused by indirect effects. As a result, the subsection entitled “Three hairpin structures in the expression platform of SAM-I_{Xcc} are important for its action in transcriptional repression” has been rewritten. The subtitle has been changed to “Three hairpin structures in the expression platform of SAM-I_{Xcc} are important for the reduction of GUS activity produced by the transcriptional fusion reporter strain responsive to SAM”. The content has been modified accordingly (page 7, line 16 to page 8, line 18 in the current version). The Supplementary Fig. 9 in the previous manuscript version has been moved from supplementary information to the main text as Fig. 2 in the revised manuscript version.

9. Page 9, line 24-25: The authors state that constructs containing all three putative hairpins demonstrated ~50% reduction in a transcriptional fusion to the reporter gene. However, Figure 1 shows a ~24% reduction and the secondary structures in Figures 1 and 2 appear to be the same, so I am led to believe that both experiments used constructs that contained all three hairpins. Why are the values different?

Response: These data sets should strictly be compared as the constructs are different. The construct pP567 shown in the previous Supplementary Fig. 9 (i. e., Fig. 2 in the revised manuscript) only contains the three hairpins (without the aptamer). However, in addition to the three hairpins, the transcriptional fusion reporter construct shown in Fig. 1 also contains the aptamer. It is not a great surprise that the two reporter strains showed different GUS activity given that they are different, although pinpointing the

specific reason for this is unclear. Moreover, it is worth noting that the reporter construct lacking the aptamer is no longer responsive to SAM (please see Fig. 2 in the current version).

10. I am also confused about the authors' interpretations of Figure 2 (page 10). The authors show that the riboswitch mediates strong repression in the presence of SAM. To go with this, they also present a reasonable secondary structure model, which predicts small base pairing segments that pair with the SD and AUG. Mutational disruption alleviates most of the repression when either half of the anti-translation helix is disrupted. When both mutations are combined, full levels of expression are observed. So, these data seem to confirm their regulatory model, yet the authors conclude the opposite. Their secondary structure model can only ever be a guess, because there are many secondary structure models that would be predicted to occur and the authors have not supported the structure model with either biochemical probing (beyond in-line probing, which is consistent with their model but does not present enough data in this region of the sequence to be conclusive) or comparative sequence alignments. That's not a criticism, just a statement. Indeed, I feel their mutational data largely support their secondary structure prediction. The authors appear to be expecting too much of a linear relationship between small helical elements and gene expression, and are ignoring all of the many other factors that will participate in the overall regulatory mechanism. Therefore, the authors must reconsider their interpretation of their own data and fully rework this section.

Response: We apologize for any confusion. We have not made contradictory conclusions based on the results presented in previous Figure 2 (i.e., the Figure 3 in the revised version). Our interpretation of the results presented in previous Figure 2 is similar to yours. To avoid confusion, we have revised this section by adding a more detailed explanation in the current manuscript version (page 9, line 16 to page 10, line 3). Additionally, to support the regulatory model proposed in the current Figure 3, the results from in-line probing experiments (current Supplementary Figures 4-7) and RNase H cleavage experiments (current Figure 5 and Supplementary Figure 16) are also used.

11. page 10, line 18-19: I was also specifically confused by the authors' interpretation of the M1+M2 mutant, where the authors state that the mutant "showed similar GUS reporter activity...indicating that translation initiation was abolished." Yet, translation levels are literally at full levels. The authors appear to be wrong in this statement.

Response: The statement is not correctly explained and has been changed in the revision to "As expected, the inhibition by SAM was completely abolished in the translational reporter strain carrying the construct with 9-nucleotide changes in the anti-AUG and anti-SD sequences (M1+M2), which in theory disrupts the sequestration hairpin and releases both the SD and AUG" in the revised manuscript (page 9, lines 9-12).

12. The authors' hypothesis that tRNA^{fMet} associates directly with the leader region RNA is bold. It should therefore be supported by solid proof. The authors present interesting evidence of intermolecular interactions between the leader RNA and the tRNA via EMSA. However, there is an inherent risk in potentially overinterpreting these data. When individual RNA molecules are mixed together at potentially high

concentrations (~10 micromolar), it is not unreasonable to expect gratuitous interactions if the RNA molecules share even short segments of base-pairing potential. Their data suggests that this pairing interaction involves nucleotides near the 3' terminus, and does not require the anticodon. The former is potentially interesting; the latter is confusing. How would the leader region avoid associating with the other cellular tRNAs if specificity determinants are not required? And, by extension, the difference in EMSA shifts for initiator and elongation tRNAs may simply result from a loss in the gratuitous base-pairing interaction due to sequence differences. The authors need to, at minimum, include experiments that involve only single stranded segments of the tRNA. It is possible that all is needed to recapitulate the EMSA shift they've observed is to include a single-stranded sequence stretch from the 3' region of the tRNA. And if that short single-stranded RNA sequence was equally capable of forming base-pairing interactions with the leader RNA, would the authors consider that meaningful, or would they then consider the possibility that it results from a gratuitous interaction? By extension, the RNase H assays also does not rule out a gratuitous base pairing interaction between tRNA and leader RNA.

Response: We agree with the suggestion that interactions resulting from normal base-pairing may take place, especially when the two RNAs are present in a relative high concentration. However, it is not difficult to distinguish such a gratuitous binding from a very specific interaction by using appropriate controls. In our case, the binding of SAM-I_{Xcc} with tRNA^{fMet} is very unlikely to be caused by gratuitous base-pairing of the two RNAs given that a single nucleotide change in the SAM-I_{Xcc} (M6 in current Figure 4b) or the tRNA^{fMet} (M_{Δ5}U) could lead to an abolishment of the binding. In addition, SAM-I_{Xcc} cannot bind with the two elongator Met-tRNAs (tRNA^{Met1} and tRNA^{Met2}) and tRNA^{fMet} cannot bind to the aptamer (please see the current Figure 4c), although there are many “short segments of base-pairing potential” in these RNA pairs. Taken together, we believe that these negative controls can completely rule out the possibility that the binding between the tRNA^{fMet} and the expression platform of SAM-I_{Xcc} is due to a gratuitous base-pairing interaction. For this reason, we think that your suggested experiment (using a short single-stranded RNA sequence in EMSA) is meaningful but is not required.

Deletion of the 3'-CCA of tRNA^{fMet} resulted in nearly lost of the binding ability but substitution of AAA for the anticodon CAU did not affect the binding (Figure 4b in the revised manuscript), indicating that 3'-CCA is involved in the binding but the anticodon is not important for the interaction. According to Reviewer #2's suggestion, we made a substitution of GGU for the 3'-CCA which did not affect the binding (Figure 4b in the revised version), indicating that the 3'-CCA sequence itself is not important for the binding interaction but the overall shape of tRNA^{fMet} is important.

According to the standard model of T-box-tRNA interaction, tRNA 3'-CCA end recognizes the UGG in the T-box loop, while tRNA anticodon recognizes the codon in the specifier loop of T-box riboswitch. It is surprising that the anticodon in the Xcc tRNA^{fMet} is not important for the binding between tRNA^{fMet} and the expression platform of SAM-I_{Xcc}. However, this may be reasonable because SAM-I_{Xcc} cannot bind with the two elongator Met-tRNAs which also contain the same anticodon as the initiator Met-tRNA (tRNA^{fMet}). It seems that the mechanism used by the SAM-I_{Xcc} expression platform to recognize tRNA^{fMet} is different from that of the known T-box

riboswitches. It is also very interesting to know how the SAM-I_{Xcc} expression platform can distinguish the initiator Met-tRNA from the elongator Met-tRNAs.

In the RNase H assay, to rule out the possibility that the increased cleavage of the SAM-I_{Xcc} expression platform RNA upon addition of tRNA^{fMet} is resulted from a gratuitous base pairing interaction between tRNA and leader RNA, another DNA oligo, which complementary to another region of the SAM-I_{Xcc} expression platform, was used as a control and the result showed that the cleavage of the SAM-I_{Xcc} expression platform RNA is not affected by addition of tRNA^{fMet} (Supplementary Fig. 15 in the current version), suggesting that the effect of tRNA^{fMet} is site-specific.

13. Page 13, line 15: correct spelling of “colectelly”. Perhaps the authors meant “collectively”. On a related note, the manuscript needs to be heavily edited for grammar and language throughout, with particular attention to the methods section.

Response: This was a typo and has been corrected. The revised manuscript has been read by a native English speaker.

14. Page 16, lines 9-10: I don’t think the authors actually mean to use “minimal inhibitory concentration (MIC)”, which is a term that means something entirely different then how the authors are using it.

Response: Apologies, we should not have used term MIC here. We have corrected in the revised version, we use a statement “a concentration that can completely inhibit...” instead (page 13, line 22 and page 14 line 5).

15. In contrast to the dogmatic language used on page 17, I’m not convinced that the experiment shown in Figure 5 alone gives enough proof to state that “these findings clearly indicate that over-expression of tRNA-fMet can derepress SAM’s inhibitory effect in vivo...”. The authors cannot rule out indirect effects from tRNA overexpression, for example.

Response: We apologize if the language use appears dogmatic, this was not the intention. The statement has been revised as “these findings suggest that...” (page 14, line 9 in the revised manuscript).

Reviewer #2(Remarks to the Author):

Review for: Novel variant riboswitch with a sensory and regulatory dual functioning expression platform The manuscript by Tang et al. presents a SAM-I riboswitch from Xanthomonas campestris which appears to have an expression platform that also acts as a sensor for uncharged tRNA^{fMet}. This is a very exciting discovery, which unfortunately the authors do not present too well in my opinion by focusing too much on the so-called “variant riboswitch”, rather than on the new tRNA sensing ability of the expression platform, the latter may be a very important contribution for amino acid biosynthesis regulation and may lead to several groups around the world looking for such motifs in other organisms. This would be one of my major comments, which would require quite a bit of re-writing. Notably, “variant riboswitch” normally is used for an aptamer domain that diverges from the consensus. I would instead put

more emphasis on what is basically a new tRNA-sensing ncRNA and the fact that it is combined in a dual sensing which requires low SAM concentration and high uncharged tRNA^{fMet} concentration for optimal expression.

Response: Thanks for your helpful suggestion. According to your suggestion, the manuscript has been revised by focusing on the tRNA sensing ability of SAM-I_{Xcc} expression platform. Please note that the previous title of the manuscript “Novel variant riboswitch with a sensory and regulatory dual-functioning expression platform” has been revised as “SAM-I riboswitch with the ability to sense and respond to uncharged initiator tRNA”.

My second major comment is that a few experiments would be more than welcome to help further validate and characterize the tRNA sensing switch. More details are provided below, together with many additional comments, in order of appearance in the main text, and then in the supplementary material. That being said, in general the work is well performed with high quality results supporting most of the author’s claims. In other words, following some re-writing and a few experiments, I recommend this paper for publication.

Response: According to your suggestion the manuscript has been revised, and several additional experiments have been performed and the results have been added into the revised manuscript. An additional figure and three additional supplementary figures have been added in the main text and the Supplementary Information, respectively. Thanks again for your helpful suggestions which improved the manuscript a lot.

Here is a non-exhaustive list of corrections and additional comments (including some important concerns that absolutely need to be looked into):

P2 Line 14: bind with uncharged -> bind uncharged

Response: This has been revised in the current manuscript (page 2, line 14).

P3 line 9: changes in response to the changes in the aptamer. - > changes in response to the stabilization of the aptamer structure.

Response: This has been revised in the current manuscript (page 3, line 9).

P3 line 18: the first sentence needs at least one reference.

Response: Reference (Ref.16) has been added in the revised version of the manuscript (page 3, line 17).

P4 line 5: was controlled -> was discovered to be controlled (or “...is in large part controlled...”, depending on the sense the authors want to convey)

Response: “was controlled” has been changed to “was discovered to be controlled” in the revised version of the manuscript (page 4, line 4).

P4 line5-6: this sentence sounds incomplete, rephrase

Response: The sentence has been revised by adding “in the control of Met biosynthesis” (page 4, lines 4-6 in the revised manuscript).

P4 line 7: as a co-activator -> as an inducer

Response: The sentence has been revised in the current version of the manuscript (page 4, line 7).

P4 lines 9-11: the authors should add the following reference: Semen A. Leyn, Inna A. Suvorova, Tatiana D. Kholina, Sofia S. Sherstneva, Pavel S. Novichkov, Mikhail S. Gelfand, Dmitry A. Rodionov (2014) Comparative Genomics of Transcriptional Regulation of Methionine Metabolism in Proteobacteria. PLoS ONE. In fact, the authors should read this paper thoroughly and amend their manuscript in many places accordingly. In particular, they may realize that the intergenic region they are studying includes a “SamR” motif (as described in Leyn et al) specific to Xanthomonas species (and perhaps also a MetR motif). This may help explain some results they discuss at the end of the manuscript, but overall does not affect significantly the conclusions of their work.

Response: Thank you for pointing this article out. Leyn et al. 2014 paper identified some potential regulatory DNA motifs (such as the SamR and MetR motif) in the promoter region of Met biosynthesis genes and some putative regulatory RNA elements in the 5'UTR of Met biosynthesis genes of Proteobacteria including Xanthomonas. The manuscript provides no functional or experimental information.

In the revised version of the manuscript, the paper by Leyn et al. (2014) has been added as a reference (Ref. 24) and the related inappropriate statements in the previous version have been revised accordingly. For example, the statement “This system of regulation in E.coli appears to be conserved in a high proportion of Gram-negative bacteria. Little information is available regarding the potential riboswitch regulation of Met biosynthesis genes in Gram-negative bacteria” (P4, lines 9-11 in the previous version) has been revised as “This system of regulation in E.coli appears to be conserved in a high proportion of Gram-negative bacteria including the Xanthomonas genus²⁴. Although potential riboswitches involved in the regulation of Met biosynthesis genes have been proposed in Gram-negative bacteria^{3,24}, none of them has been functionally characterized” in the revised manuscript (page 4, lines 8-12).

Please note that a putative MetR box (motif) is indeed present in the promoter of Xcc met operon. However, we do not discuss this aspect in this manuscript because the results from the study on the XC1251 promoter was published previously (Zhang et al. 2018. The Gram-negative phytopathogen Xanthomonas campestris pv. campestris employs a 5'UTR as a feedback controller to regulate methionine biosynthesis. Microbiology 164, 1146-1155) (Ref.26 in the revised version). In the publication, we demonstrated that the promoter of XC1251 is not involved in Met-responsive regulation of the met operon.

As soon as the gene XC1251 is mentioned, its function should be mentioned in the main text (homoserine O-acetyltransferase).

Response: The product of XC1251, homoserine O-acetyltransferase is mentioned in the first place it appears (the legend of Figure 1).

P4 line 13: there are actually many SAM riboswitches in Gram negative bacteria, including SAM-II, SAM-V and SAM-SAH (although these only regulate metK, Weinberg et al. 2010, Genome Biol). This information can be found in Rfam, including the SAM riboswitch described in this paper (which is a reason why calling

it “a novel variant” seems inappropriate). Because of this, I suggest the authors double-check to see if indeed no other SAM riboswitches were functionally characterized in Gram negatives before (although it would be simpler to delete this statement, since it is not the most important contribution of this work in any case).

Response: Thank you for pointing out this issue. We have checked the literature carefully and make sure that no SAM-I riboswitch has been functionally characterized in Gram-negative bacteria so far, although some potential SAM-I riboswitches have been predicted in Gram negative bacteria. The terms “a novel variant” and “variant riboswitch” are indeed inappropriate for SAM-I_{Xcc}. In the revised manuscript, we use “a new type of riboswitch” instead.

P4 line 16: provided the first example of a -> provided evidence of

Response: This has been revised in the current manuscript (page 4, line 15).

P4 line 18: of this amino acid -> of Met

Response: This has been revised in the current manuscript (page 4, line 17).

P4 line 19: importance -> concern

Response: This has been revised in the current manuscript (page 4, line 17).

P4 line 20: It is not clear what “Despite this observation” refers to given the text structure (eventhough we can easily guess). The sentence could simply start with “The mechanism...”

Response: This has been revised in the current manuscript (page 4, line 18).

P4 line21: demonstrating this -> demonstrating that this

Response: This has been revised in the current manuscript (page 4, line 20).

P4 line 21-22: As stated before, this SAM riboswitch (the aptamer domain) was already predicted and could be found in Rfam’s annotations, so saying that it has “structural similarities to the SAM-I riboswitch class” is an understatement, it was already predicted through homology searches as a SAM-I.

Response: In the current version, the previous statement “Here, we provide evidence demonstrating this 5’UTR region from Xcc encodes a riboswitch with structural similarities to the SAM-I riboswitch class from Gram-positive bacteria” has been revised as “Here, we provide evidence demonstrating that this 5’UTR region from Xcc encodes a functional SAM-I riboswitch” (page 4, lines 19-21 in the revised manuscript).

P4 line 23: “Our approach...”, add some details, just a few words to make the sentence and statement more precise.

Response: This has been revised in the current manuscript (page 4, line 21).

P5 line 4: platform functions -> platform also functions

Response: This has been revised in the current manuscript (page 5, line 2).

P5, end of introduction: here would be one of the good places to better emphasize the findings regarding the new tRNA-sensing RNA.

Response: The statement at the end of the introduction has been revised to say “The findings describe a structurally typical SAM-I riboswitch from *Xcc* with a previously uncharacterized mode of action. SAM-I_{*Xcc*} appears to be broadly distributed in Gram-negative *Xanthomonas* species bacteria and its expression platform represents a novel natural tRNA-sensing RNA element.” (pag 5, lines 2-6 in the revised manuscript).

P5 line 13: the amino acid -> Met

Response: This has been amended in the revised manuscript (page 5, line 12).

*P5 line 17-18: authors state that no sequence similarity to any known expression platform was found. How have the authors verified this? Except for a comparison with *Bacillus subtilis*, there is no sequence alignment in this manuscript, I will come back to this later.*

Response: Reviewer #1 also pointed this out. We removed the statement on page 5, lines 17-18 in the previous manuscript, i.e., “However, the putative expression platform encoded by SAM-I_{*Xcc*} exhibits no sequence similarity to any known SAM-I riboswitches or any other characterized riboswitches” from the text in the revised manuscript. Please see above for further details.

P5 line 19-22: even if one characterized example of a given riboswitch class has a transcription terminator-based expression platform, it is not assumed that all other riboswitches of that same class have the same type of expression platforms, the latter tend to vary more than aptamer domains. Authors may wish to refer to ref 24 in that regard. Also, transcription termination can also occur through a Rho-dependant pathway, more common in gram-negatives (the “consensus” motifs recognized by Rho are C-rich, which would fit with the loops of P5, P6 and P7 in the model provided in Fig1).

Response: The statement on page 5, lines 19-22 in the previous version has been revised (page 5, lines 15-20 in the revised version). In addition, the other reviewer pointed out that the conclusion that SAM-I_{*Xcc*} also control gene expression at the transcriptional level (page 6, lines 11-14 in the previous version) is inappropriate (reviewer #1’s Question #6 & #8), because we cannot rule out the possibility that the SAM-responsive GUS activity repression may be caused by other indirect effects. The subsection entitled “Three hairpin structures in the expression platform of SAM-I_{*Xcc*} are important for its action in transcriptional repression” has been rewritten (page 7, line 16 to page 9, line 18 in the revised manuscript). The previous Supplementary Fig. 9 has been moved from Supplementary Information to the main text (Figure 2 in the revised manuscript).

P6 line 2: known bacteria -> known that bacteria

Response: This has been revised according to your suggestion in the revised manuscript (page 5, line 25).

P6 line 3: the reference 29 relates to yeast, which forces us to reconsider the statement of that sentence. Another reference should be used.

Response: The original reference 29 has been replaced by an appropriate reference (Ref.29) in the revised manuscript (page 6, line 1).

P6 line 5-6 and 8: the exact GUS expression numbers are provided in the legend, it is redundant in the text (but % variation can be kept).

Response: The exact GUS expression numbers in the text have been removed in the revised manuscript (page 6, lines 3-6).

P6 line 10: the alternative amino acid glycine at a concentration of 0 uM or 300 uM - >(5 corrections) an alternative amino acid, glycine, at a concentration of 300 uM

Response: This has been amended in the current version of the manuscript (page 6, lines 7-8).

p7 lines 8-9: again, this prediction was already found in Rfam.

Response: The statement “The overall architecture of the SAM-I_{Xcc} aptamer is nearly identical to that of known SAM-I aptamers (Fig. 1c, Supplementary Fig. 7). Modeling of SAM-I_{Xcc} shows features identical to SAM-I rather than SAM-IV riboswitch including a P4 hairpin in the core, a lack of an additional pseudoknot, a kink-turn in the P2 stem and uridine residue at position 121” (page 7, lines 8-12 in the previous version) has been revised as “The overall architecture of the SAM-I_{Xcc} aptamer is nearly identical to SAM-I aptamer rather than SAM-IV aptamer, including a P4 hairpin in the core, a lack of an additional pseudoknot, a kink-turn in the P2 stem and uridine residue at position 121 (Figure 1c, Supplementary Figure 8)” (page 7, line 9-12 in the revised version).

P9 line 8: D – (d)

Response: This has been revised in the current version of the manuscript (page 29, line 22).

P9 line 12: There is no description of the modeling (neither in the results, nor in the methods, not even in supplementary material). Some brief description should be provided, even if it was performed manually by some of the authors.

Response: This section has been rewritten in the revised manuscript (page 7, line 16 to page 8, line 18). See above for further details.

P9 line 17: constructs are introduced -> constructs were introduced

Response: This has been revised in the current version of the manuscript (page 7, line 25).

P9 line 18: -> without SAM supplementation.

Response: This has been revised in the current version of the manuscript (page 8, line 2).

P9 line 23 and 26-27: the way the text is written, it implies that SAM downregulates transcription, but this is not the case as shown by the authors in suppl fig 9 (the +/- SAM pairs always have the same letter, meaning there is no significant difference between each). Even if some mutants do seem to affect transcription. The text should

be changed to ensure this is clear. Also, the notation of the GUS constructs with regards to translational vs transcriptional fusions are not always clear (they should have the SD+ or SD- label, as in the text).

Response: Please note that the original Supplementary Fig. 9 has been moved from Supplementary Information into the main text as Figure 2 in the revised version of the manuscript. The fact that the GUS activity of the reporter strains shown in this figure was affected upon addition of SAM is due to the lack of SAM-binding domain (the aptamer) in the constructs. According to the proposed regulatory model of SAM- I_{Xcc} , the expression platform will be in a permanent OFF state without the aptamer regardless of the presence or absence of SAM. All of the GUS constructs shown in the Figure are transcriptional fusion, which has been indicated more clearly in the revised manuscript.

P10 line 4: add (Fig. 1) at end of sentence.

Response: This has been revised in the current version of the manuscript (page 8, line 22).

P10 line 4: -> The potential mechanism by which SAM- I_{Xcc} inhibits...

Response: This has been revised in the current version of the manuscript (page 8, line 22).

P10 line 4-5: -> that inhibit translation initiation. The in-line probing experiments revealed...

Response: This has been revised in the current version of the manuscript (page 8, lines 23-24).

P10 line 18: and release -> and releases

Response: This has been revised in the current version of the manuscript (page 9, line 12).

P10 line 18: I assume that instead of “wild-type”, the authors mean “with or without SAM”, also, the following text would be better as “... indicating SAM-mediated modulation of translational initiation was abolished”

Response: The sentence “A mutant that carries 9-nucleotide changes to SAM- I_{Xcc} (M1+M2), and in theory disrupts the sequestration hairpin and release both the SD and AUG, showed similar GUS reporter activity to wild-type, indicating that translational initiation was abolished (Fig. 2c)” (page 10, lines 16-19 in the previous version) was revised as “As expected, the inhibition by SAM was completely abolished in the translational reporter strain carrying the construct with 9-nucleotide changes in the anti-AUG and anti-SD sequences (M1+M2), which in theory disrupts the sequestration hairpin and releases both the SD and AUG (Fig. 3c)” (page 9, line 9-12 in the current version).

P10 line 19-20: mutant that carries changes that expose the SD -> mutant that that exposes the SD P10 line 22: -> construct that exposes the start codon AUG...

Response: These have been revised in the current manuscript (page 9, lines 12-15).

P11 line 1: mediation -> modulation (or repression?) It is not clear what the authors mean here, in part because according to the models in figure 2, the SD is still sequestered, even if P11, figure 2a: add the orange highlight (for anti-AUG) to the “SAM –” version of the structure.

Response: According to your suggestion and the other reviewer’s comment, we rewrote this section (page 9, lines 16 to page 10, line 2 in the revised manuscript). The anti-AUG has been highlighted in the SAM- version of the structure.

P12 line 4-5: change sentence to: “The homoserine O- acetyltransferase is connected to Xcc Met metabolism pathway and, by extension, to uncharged/charged Met-tRNA.

Response: This has been revised according to your suggestion in the current version of the manuscript (page 10, lines 7-9).

P12 line 6-7: was too interacted with -> was to interact with

Response: This has been amended in the current version of the manuscript (page 10, line 10).

P12 line 7: method -> mechanism

Response: This has been amended in the current version of the manuscript (page 10, line 11).

P12 line 8: have several features -> have features

Response: This has been amended in the current version of the manuscript (page 10, line 11).

P12 line 10: can interact Met-tRNA -> can interact with Met-tRNA

Response: This has been amended in the current version of the manuscript (page 10, line 13).

P12 line 15: was -> were

Response: This has been amended in the current version of the manuscript (page 10, line 19).

P12 line 17: -> Additionally, the aptamer domain and the expression...

Response: This has been amended in the current version of the manuscript (page 10, lines 21-22).

P13 line 15: Colectelly -> Collectively

Response: This has been amended in the current version of the manuscript (page 12, line 12).

P13 line 16: While several mutants were analyzed for the anti-SD, none were analyzed for tRNA binding in the anti-AUG (P7a). This is surprising, given that such a mutant has been tested (Fig. 2 for SAM-mediated regulation of GUS activity). It would also seem particularly interesting to do such a mutant given that this short sequence encompasses a “UGG” sequence, which would be an ideal target for the

tRNA-CCA. This would also allow to perform a complementation with a corresponding mutant tRNA. Such an experiment would provide additional support to the tRNA-sensing RNA region the authors uncovered, as well as provide a more detailed model of how this interaction can occur. Clearly, a lot of work would be required to understand in details how this RNA structure distinguishes tRNA-fMet from other tRNAs, but this presumably simple experiment would still be a great help to establish a “basic model” that may serve as the foundation of further research.

Response: We carried out additional experiments and tested the effect of several mutations in tRNA^{fMet} (including substitution of GGU for the 3'-CCA, deletion of the first U at the 5'-end, mutations in D-loop and T-loop) and in the SAM-I_{Xcc} expression platform (mutations in the anti-AUG sequence and the CCA complementary sequence UGG) on the binding ability, and got some interesting findings. These results have been shown in Figure 4 and Supplementary Figure 12 and discussed in the text of the revised manuscript (page 11, line 12 to page 12, line 22).

P14 line 8: -> methionine-tRNAs

Response: This has been amended in the current version of the manuscript (page 31, line 9).

P14 line 10: were -> are

Response: This has been amended in the current version of the manuscript (page 31, line 11).

P14 line 11: were -> are

Response: This has been amended in the current version of the manuscript (page 31, line 12).

P16 line 10: “MICs” usually express growth inhibition, to prevent confusion, authors are invited to rephrase

Response: The phrase “minimal inhibitory concentration (MIC)” used here is inappropriate, which has been revised as “a concentration that can completely inhibit ...” in the revised manuscript (page 13, line 22).

P16 line 17: when saying “the MIC condition of SAM”, it is not clear what the authors refer to, two concentrations have been used. Maybe what they mean is “at the corresponding MIC condition”. Note that the use of MICs here is a bit unusual. In the past, similar riboswitch studies have been performed in conditions where there is an excess of the ligand, why the authors have not used such an excess? Also, if they have, it would be interesting to know what were the results, perhaps this was to have a more “borderline” expression level that could more easily “switch” according to tRNA expression? If it is the case, it would be interesting to know and it would actually add to the interest of the story by providing further evidence of the complex and subtle interplay between different sensing mechanisms.

Response: As detailed above, in the previous version of the manuscript the phrase “MIC” was improperly used, which has been revised in the current version. In the experiment, a series of SAM concentrations (0, 25, 50, 100, 150, 200, 250, 300 μ M) were used (Supplementary Figure 19a). At 200 μ M concentration the GUS activity of

the translational reporter strain was significantly reduced and at 250 μM or 300 μM the GUS activity was completely inhibited (Supplementary Figure 19a). The concentration 250 μM or 300 μM may be an excess one. According to your comment, the statement “cultured at the MIC condition of SAM” has been revised as “cultured in the minimal medium supplemented with 250 μM SAM, a concentration that can completely inhibit the GUS activity of the translational reporter strain” (page 14, line 5).

P17 line 16: -> can partially derepress

Response: This has been amended in the current version of the manuscript (page 14, line 9).

P17 line 20: mediation -> regulation

Response: This has been amended in the current version of the manuscript (page 14, line 13).

P19 line 2: as mentioned, the claim of “we identified and characterized a variant riboswitch” should be amended, both because of the “identified” (the SAM riboswitch was already predicted by Rfam) and the term “variant riboswitch”. In contrast, the characterization is obviously very significant.

Response: We have revised these statements as “we characterized the riboswitch SAM-I_{Xcc} that regulates...” in the revised manuscript (page 14, line 22).

Note that from line 4 to 13 there is some redundancy and some sentences could consequently be simplified.

Response: This section has been revised (page 14, line 23 to page 15, line 7).

P19 line 4: We detail -> We performed

Response: This has been amended in the current version of the manuscript (page 14, line 23).

P19 line 6: -> we also demonstrate

Response: This has been amended in the current version of the manuscript (page 15, line 1).

P19 line 7: delete “whose”

Response: This has been amended in the current version of the manuscript (page 15, line 1).

P19 line 7: with previously uncharacterized dual ability. -> with a dual sensing ability.

Response: This has been amended in the current version of the manuscript (page 15, line 2).

P19 line 11-13: it would be good to look if what the authors found is the first riboswitch expression platform which also has a sensing function, if so, a statement

could be made regarding that (and even if there are other examples, it will likely be one among very few such regulatory regions)

Response: The SAM-I_{Xcc} expression platform is indeed the first riboswitch expression platform found to also have a sensing function. A statement, “As far as we know, the expression platform of SAM-I_{Xcc} is the first riboswitch expression platform validated to have sensing function”, has been added in the revised manuscript (page 15, lines 6-7).

P19 line 16-19, sentences are not clear and require references

Response: This section has been amended in the revised manuscript (page 15, lines 10-15).

P19 line 20-22: what the authors describe as “variants” are rather different combinations of riboswitches in the same UTR, which is also what their manuscript describes, in addition to describing a new type of tRNA-sensor.

Response: This section has been amended in the revised manuscript (page 15, lines 15-17).

P19 line 23: This -> The

Response: This has been amended in the revised manuscript (page 15, line 17).

P19 line 24: change phrasing to avoid the term “variant”

Response: The use of the term variant has been amended in the revised manuscript (page 15, lines 15-17).

P20 line 15: “more accurate”, perhaps “more thorough sensing” would be more appropriate?

Response: This has been amended in the revised manuscript (page 16, line 9).

P20 line 25: t-box elements have been predicted in proteobacteria before, so this is not the first example of a tRNA-sensing element in Gram negative bacteria. However, it is, to my knowledge, the only RNA capable of sensing tRNAs outside of t-box elements, which is a very exciting finding (and the fact that it is found in the expression platform of a SAM riboswitch is a bonus to make it even more interesting).

Response: The putative T-box elements have been predicted in some Gram-negative bacteria; however, none of them has been functionally characterized so far. SAM-I_{Xcc} is the first functionally characterized tRNA-sensing riboswitch in Gram-negative bacteria. The statement “As far as we know, SAM-I_{Xcc} is the first tRNA-sensing riboswitch identified in Gram-negative bacteria” has been revised as “As far as we know, SAM-I_{Xcc} is the first RNA found to be capable of sensing tRNAs outside of T-box elements” in the current version (page 16, lines 19-20).

P21 line 4: -> For T-box, sequence complementarity...

Response: This has been revised in the current manuscript (page 16, lines 23-24).

P21 line 6: box loop; between the anticodon...

Response: This has been revised in the current manuscript (page 16, line 25).

P21 line 6: delete existed

Response: This has been revised in the current manuscript (page 16, line 25).

P21 line 7, delete “and fitness”

Response: This has been revised in the current manuscript (page 17, line 1).

P21 line 7, partners are essential -> partners have been shown to be essential

Response: This has been revised in the current manuscript (page 17, line 1).

P21, Line 9: complementarity and overall shape fitness are -> and overall shape complementarity are

Response: This has been revised in the current manuscript (page 17, line 3).

P21 line 10: unlike other -> unlike the other

Response: This has been revised in the current manuscript (page 17, line 4).

P21 line 13-15 rephrase sentence

Response: This has been revised in the current manuscript (page 17, lines 8-9).

P21 line 18: the term “widespread” does not seem appropriate, given that it is apparently only found in Xanthomonas. It would also be interesting to know if it is found in all Xanthomonas species and only upstream of this gene, or also genes encoding other activities (it is of interest that Xcc encodes two homoserine O-acetyltransferases).

Response: This has been revised in the current manuscript. The statement “SAM-I_{Xcc} homologues are widespread and highly conserved in the 5’UTR of *metA*, a gene encoding the key Met biosynthesis enzyme homoserine O-acetyltransferase, in many *Xanthomonas* species” has been revised as “SAM-I_{Xcc} homologues exist and are highly conserved (more than 90% identity) in the 5’UTR of *metA*, a gene encoding the key Met biosynthesis enzyme homoserine O-acetyltransferase, in nearly all *Xanthomonas* species whose genomes have been sequenced” in the revised manuscript (page 17, lines 12-15).

As you mentioned, *Xcc* encodes two homoserine O-acetyltransferases. In the genome of *Xcc* strain 8004, the two genes are named *XC1251* and *XC1889*. Sequence analysis revealed that SAM-I homologue only exists in the 5’UTR region of *XC1251* but not *XC1889* or other *met* genes.

P21 line 21: More importantly -> Moreover

Reply: This has been revised in the current version of the manuscript (page 17, line 16).

With regards to conservation, this paper lacks a lot of details. An alignment should at least be presented in supplementary material, especially given that authors claim the

motif is conserved. I have made such an alignment following a few BLAST searches, here is a summarized alignment with non-redundant sequences

Response: Thank you very much for making an alignment for us. A full list of the distributions of SAM-I_{Xcc} homologues in sequenced *Xanthomonas* and other bacteria has been published previously (Zhang et al., 2018, i.e., the reference 26 in the current version of the manuscript). The SAM-I_{Xcc} homologues among *Xanthomonas* species are highly conserved with an identity of more than 90%. This information has been added into the revised manuscript (page 17, lines 12-15).

P22 line 15: MgCl2 (subscript)

Response: This has been amended in the current manuscript (page 18, line 11).

P22 line 25: no data were presented for in-line probing at room temperature, this might be interesting to have to compare patterns, especially given that Xcc does not normally grow at 37C as would a human pathogen. Also, rather than GC content, the reason why it was difficult to get modulation at room temperature might be that the full expression platform is tested together with the aptamer. Often aptamers are tested alone, when tested with their expression platform it is not uncommon to lose modulation as the binding of ligand is often co-transcriptional, meaning that the ligand has the possibility of binding even before the expression platform is fully transcribed.

Response: This has been revised in the current manuscript. We have added the in-line probing assay at room temperature (22 °C) and 37 °C (page 27, lines 1-2 in the revised manuscript) in the supplementary data (See supplementary figure 3). As suggested, we performed the in-line probing with the aptamer only and the full-length of the riboswitch (including the expression platform). In comparing, it is clear that at room temperature barely any modulation could be seen no matter with the aptamer only or the full-length. As temperature rose to 37 °C, modulations were observed for both constructs (See supplementary figure 3). Based on these findings, we speculate that the difficulty to get modulation *in vitro* at room temperature is very likely due to the high GC content.

P23 gus reporter constructs: more details would be welcome in suppl Fig. 2 (the sequence from promoter up to the first few codons of gus; for both the SD+ and SD- constructs).

Response: This has been revised in the current manuscript. Specifically, supplementary Figure 2 has been revised to include this detail.

P24 line 10: lysised -> lysed

Response: This has been amended in the current manuscript (page 20, line 6).

P24 line 18: milligrams -> milligram

Response: This has been revised in the current manuscript (page 20, line 14).

P25: I have never seen "none-labeled", unlabeled

Response: This has been amended in the current manuscript (page 20, line 24).

P26 line15: in the PBS -> in PBS

Response: This has been revised in the current manuscript (page 22, line 11).

P26 line 19: was -> were

Response: This has been revised in the current manuscript (page 22, line 15).

P26 line 21: the -> a

Response: This has been revised in the current manuscript (page 22, line 17).

P27 line 3: the -> a

Response: This has been revised in the current manuscript (page 22, line 23).

P27 line 18: author -> authors

Response: This has been revised in the current manuscript (page 23, line 13).

Supplementary material:

Most importantly, some info with regards to sequence conservation should be provided (and at least some mention to what extent it supports the structure model proposed). Figures of the two other tRNA-Met tested should be presented

Response: As discussed above, a full list of the distributions of SAM-I_{Xcc} homologues in sequenced *Xanthomonas* and other bacteria has been published previously (Zhang et al., 2018, i.e., the reference 26 in the current version of the manuscript). The SAM-I_{Xcc} homologues among *Xanthomonas* species are highly conserved with an identity of more than 90%. This information has been added into the revised manuscript (page 17, lines 13-15). Additionally, the detailed information (including the sequence, genome location and the predicted secondary structure) of the two other tRNA-Met has been added in the revised manuscript (Supplementary Figure 12).

S3 line 5: aend -> end

Response: This typo has been corrected in the revised manuscript.

S6: it is not possible to determine a Kd below ~ 1 nM by in-line probing because at lower concentrations, there is more RNA than ligand molecules, which would falsely give the impression that the RNA is not modulating (as compared to absence of ligand). Therefore, the concentration of 10 pM, and even 100 pM are not very useful.

Response: We have kept the in-line probing data from the 10 pM and 100 pM SAM in Supplementary Fig 7, but removed data derived from the two concentrations for Kd determination (see Figure 1e).

P13-14 this is strangely organized and could not find where it fits in the main text, reorganize P15 line 5: gnome -> genome

Response: The primers shown on P13-14 in the previous version of the manuscript have been moved to the primer table, i.e., Supplementary Table 2 in the revised version. The typo “gnome” has been corrected.

P15 line 8: red -> purple

Response: This has been amended in the revised manuscript (page 13, line 9).

P15, the reference Leyn et al 2014 should probably be used to improve this figure. (or at least be cited)

Response: The reference Leyn et al (2014) has been added in the figure legend of Supplementary Fig. 10 in the revised manuscript (page 13, line 5).

Fig S11: part (A) of this figure is taken integrally from Gutiérrez-Preciado et al. (2009) Biochemical Features and Functional Implications of the RNA-Based T-Box Regulatory Mechanism. MICROBIOLOGY AND MOLECULAR BIOLOGY REVIEWS. Unless authors had consent from the journal and authors, this should be taken out of their manuscript (or they should ask for the permission and, if they get it, indicate that the figure was taken from that article, and not merely cite it).

Response: We have revised the figure by including an element of the figure presented by Gutiérrez-Preciado et al. (2009).

P22: other factors could explain the difference in regulation, for instance, the plasmid is multicopy. Also, authors might want to discuss the presence of a conserved “Xanthomonadales” box in the promoter region of XC1251, named SamR by (Leyn et al 2014).

Response: Thanks for your suggestion. An explanation for the difference in the inhibitory concentration of SAM in the GUS reporter strain and the FLAG-tagging strain has been revised in the current version of the manuscript (page 23, line 11 to page 24, line 1 in the Supplementary Information). However, we decided to not discuss the conserved “SamR” box associated with the promoter of *XC1251* but just point out that this has been detailed in a recent publication by the team (Zhang et al., 2018, i.e., the reference 26 in this manuscript).

Reviewers' comments:

Reviewer #2 (Remarks to the Author):

Second Review for :

Novel variant riboswitch with a sensory and regulatory dual functioning expression platform
Tang et al. have provided a rebuttal which answers many of my concerns (apparently as well as for reviewer 1). I still think that this is a very interesting manuscript with big potential for impact. However, there are still many minor points that would require revision, as well as a few potentially more important aspects. For this second revision, I have gone through both the rebuttal to my previous comments and to those of reviewer 1.

In order of appearance in the rebuttal and/or text:

1-

The following should be added to the amended text in p6, lines 8-14 of the current version:

"... or that this effect may be due to indirect effects of SAM on transcription in general."

2-

Perhaps the biggest comment I still have to make concerns the "new type of t-box" reported in this manuscript. I agree with reviewer 1 that this is a bold claim and that, while it is supported by a few lines of evidence in the first version of the manuscript, further support would be welcome to make sure the proposed model is right for the "big picture" (even if some details, such as the exact structure of P5,P6,P7, might differ from the current prediction). The revised version includes a few such experiments, such as the RNase H proings and changing CCA to GGU (which surprisingly does not change the ability of the tRNA to bind the expression platform), as well as D-loop and T-loop mutants. With all the mutations assayed, the only ones that help define the proposed specificity for the initiation tRNA-fMet are the 5'U and UA just upstream of the CCA, all the other mutations that affect binding to the expression platform are in features common to all tRNAs (D-loop, T-loop, deleted CCA or added 3'C). I noted the possibility of 9 bp between the two RNAs, which actually fits with results of expression platform mutants. The M2 mutation changes AUCC to UAGG, completely abolishing the capacity of this RNA to interact with tRNA^{fmet}. This sequence could base pair with tRNA, actually, there are 9 consecutive bases that can do watson-crick bp with the tRNA first junction and 5' end (CGGGGUGGA of tRNA can bp with UCCACCCCG of stem 7, which also fits with a loop of that structure, so presumably more accessible). While less drastic, the M6 would disrupt one of bp of that interaction, perhaps this would suffice in preventing it. M7, would behave as M2.

The dilemma is to either properly support the "new type of t-box", which corresponds to an important discovery of a novel sensing non-coding RNA structure which would be a high impact finding, or tune down this finding and simply present this as a potential way by which the riboswitch expression platform may be somewhat affected. For the former, several lines of evidence are already presented, but additional controls could further help dissipate doubts regarding the model. For instance, for Fig6, a negative control where one of the expression platform mutants known not to interact with the tRNA (like 8004/pM2-SD-) is tested in the WT strain or those overexpressing the tRNAs (especially tRNA-fMet)

would dissipate doubt regarding a potential general effect on translation of overexpressing the initiation tRNA (although, admittedly, the RpoB control seems rather constant). As for in-vitro tests, it is relatively common to include competitors, especially given the fact that at 10 uM, non-specific interactions might not be so surprising. Regarding this, in their rebuttal, authors mention that the base pairing is unlikely to be gratuitous because some single point mutations suffice to prevent the interaction, but if the fortuitous interaction happens to occur through base-pairing in that region (or through base-pairing with bases which availability is affected by that region), it could have such an impact as well. In other words, while it does seem like a specific interaction, we cannot completely rule out a relatively non-specific interaction. Another experiment that could help in that regard is determining the K_d, which would be possible with the EMSA assays the authors have used. That being said, given that the range of cellular concentrations of tRNAs, which just so happens to be in the range of ~2 to 10 uM for methionine tRNAs in E. coli

([https://bionumbers.hms.harvard.edu/files/The%20intracellular%20concentration%20\(%C2%B5M\)%20of%20tRNA%20isoacceptors%20as%20a%20function%20of%20growth%20rate.pdf](https://bionumbers.hms.harvard.edu/files/The%20intracellular%20concentration%20(%C2%B5M)%20of%20tRNA%20isoacceptors%20as%20a%20function%20of%20growth%20rate.pdf)), thus suggesting these concentrations are not so high after all.

3-

P7 line 18 to P8 line 16:

While the authors did change several aspects of the text, I am not convinced this section is absolutely necessary in the main text, the effect of the stems on transcription (or even RNA stability) is fairly small if the proposed stems are really involved in such a regulatory mechanism. I would argue that this whole section could be summarized in ~2 sentences briefly describing how deleting each stem individually adds no significant effect on RNA levels, and that two or three of the stems needed to be deleted to have an effect of ~2 fold increase compared to the three stems found in WT.

4-

If authors do keep that section (or wish to move it to supplementary material), here are some minor corrections:

P7line21: the decrease of -> the decrease of

Line 22: To confirme -> To confirm

P8 line 3: lacking -> lack

Line 9: bythe -> by the

5-

Page9 line 21:

Authors mention that GUS activities in absence of SAM "are very similar" in Fig 3c, actually, they are identical. Clearly, each of the constructs was normalized to its "no SAM" condition. So this statement can't be deduced from fig 3c, we would need un-normalized data to say this. The text needs to be changed and/or data presented differently.

6-

Page9 line 23: explantation -> explanation

Lin25: is existent -> exists

Also on line 25, "Fig. 5b" is mentioned before "fig 4" and suppl fig 16 before many other suppl figures, although this does not cause problems to the reader, it is somewhat unusual not to number figures in order of appearance.

7-

Page 17 line 10:

The statement "a new type of riboswitch", while better than the previous version, might still be better if changed the more precise statement: "a riboswitch with a new type of dual sensing capacity"

8-

Page 21 line 12: none-labeled -> unlabeled

9-

Page 30 line 4: insersion -> insertion

10-

The following comments/correction follow the numbering of the second version of the manuscript at the end of the file.

Line 299: presumably authors meant GGU instead of UUG.

Line 311: Furtheromre -> Furthermore

11-

Regarding the rebuttal for the reference of SAM transporters, while the new reference does refer to bacteria SAM transport, it is a very specific case. The abstract of that reference actually states:

"We have confirmed the presence of an AdoMet transporter in the rickettsiae which, to our knowledge, is the first bacterial AdoMet transporter identified."

So it actually argues against specific transport of SAM and, browsing through the article, I could not find anything related to non-specific transport of SAM through amino acid permease.

Many important experiments of this manuscript rely on SAM supplementation in the medium, but even if there are no known SAM transporters, some non-specific transport could presumably occur (either of SAM itself or of SAM breakdown products, such as methionine). The authors' results clearly show riboswitch-mediated regulation in presence of SAM, which is completely in accordance with what would be expected if intracellular SAM concentrations increased in these conditions. In other words, this does not jeopardize the whole manuscript, but the statement still remains inappropriate in its current format.

12-

Line 152:

for the proposed changes, instead of "...is nearly identical to SAM-I aptamer..." I would rather suggest: "... corresponds to SAM-I aptamer..."

13-

To clarify the author's rebuttal, in their response to my concerns with supplementary figure 9 (now figure 2) (p11-12 of the rebuttal pdf), authors' mention that the GUS activity was affected by SAM addition, I assume they meant "was not affected", otherwise I disagree with their interpretation.

14-

It would be interesting to mention somewhere in the manuscript that Xcc encodes two homoserine O-acetyltransferases, as this may be meaningful regarding the biology of the regulation of this gene.

15-

Regarding the addition of an alignment, while Xhang et al. 2018 present an alignment of the UTR, it was not described as an RNA structure in this manuscript, as such it would still be beneficial to include an alignment to allow readers to judge the conservation of the motif with proper format to evaluate the structures described (at least in supplementary material).

As a helper:

```
Query 1 ACGCGAGCTCCCGGAAG.CTCGATGGCCGATCCACC..CCGGATATCGCCATG 51
CP000050.1 1531183 ACGCGAGCTCCCGGAAG.CTCGATGGCCGATCCACC..CCGGATATCGCCATG 1531233
CP016830.1 1274832 ACGCGAGCTCCCGGAAG.CTCGATGGCCGATCCACC..CTGGACATCGCCATG 1274882
CP018728.1 4131786 ACGCGAGCTCC-GCGAAG.CTCGATGGCCGATCCACC..CCGGATACCGCCATG 4131737
CP019515.1 3062879 ACGCGAGCTCC-GCGAAG.CTCGATGGCCGATCCACC..CTGGATACCGCCATG 3062830
CP020971.1 4621205 ACGCGAGCTCC-GCGAAG.CTCGATGGCCGATCCACC..CTGGATACCGCCATG 4621254
CP034653.1 1907635 ACGCGAGCTCC-GCGAAG.CTCGATGGCCGATCCACC..CTGGATACCGCCATG 1907684
CP020987.1 934400 ACGCGAGCTCC-GCGAAG.CTCGATGGCCGATCCACC..CTGGATACCGCCATG 934351
CP013004.1 1576411 ACGCGAGCTCC-GCGAAG.CTCGATGGCCGATCCACC..CTGGATACCGCCATG 1576460
CP022270.1 3461809 ACGCGAGCCCC-GCGAAG.CTCGATGGCCGATCCACC..CTGGATACCGCCATG 3461760
CP013679.1 1768456 ACGCGAGCTCC-GCGAAG.CTCGATGGCCGATCCACC..CTGGATACCGCCATG 1768407
CP019797.1 2553242 CCCGGAAG.CTCGATGGCTGCTCCTTA..CCGGATATCGCCATG 2553201
CP033586.1 1002918 ACGCGAGC-CCCGGAAG.CTCGATGGCCGATTCCCC..CCGGATCCCGCCATG 1002970
LS483406.1 3498410 ACGCGAGC-CCCGGAAGgCTCGATGGCCGATTCCCCaCCGGATCCCGCCATG 3498358
..... "*"*****" /"***** " * * * * * * * * * *
```

(unfortunately, colors don't come out in this format, hopefully the authors got the correct format the first time)

16-

Regarding the figure taken from another publication, I suggest the authors properly review the copyrights and each journal details regarding permissions required for inclusion of even portions of a figure in another article. Also, while the correct journals are cited, authors should add something such as "adapted from" or "taken from" in the legend of the figures burrowed from other articles. Looking back at some figures, I also noted that suppl fig 8 (a) and (b) come from the cited reference and that (d) and

(e) come from the other corresponding cited reference. So, even though this was likely included in good faith, some verifications or permissions are required.

Reviewer #3 (Remarks to the Author):

In this manuscript, Tang et al. report a novel riboswitch from *Xanthomonas campestris* (Xcc). This riboswitch recognizes S-adenosyl methionine (SAM), and in doing so appears to function similar to a classical SAM riboswitch – by changing conformation in the presence of SAM, resulting in altered gene expression. Indeed, the SAM-IXcc switch was identified in part through a homology search. However, this riboswitch has the unexpected and novel activity of also binding to and responding to uncharged initiator Met tRNA. The authors demonstrate ligand-dependent conformational changes (in response to SAM) by in-line probing. These effects while convincing are somewhat subtle and temperature-dependent: the authors posit that this is due to high GC-content or due to the modest K_d of SAM, though it could also be that in this unusual system the aptamer behaves somewhat differently in isolation than in the context of the expression platform. Further, the authors report in an expression system that (1) the effects of the switch occur at the translational level, (2) three stem loops in the expression platform are critical to activity (one of which occludes the Shine-Delgarno sequence), (3) the Met tRNA binds to one of these hairpins, and in doing so influences translation activity.

This paper has undergone extensive revision in response to a previous round of reviews. The revised manuscript is clearly written, and the authors have responded to all reviewer comments. I agree strongly with the previous reviewers that the key points in this manuscript are related to the tRNA-sensing aspects of this riboswitch. These discoveries are novel and warrant publication in *Nature Communications* for two reasons – (1) this is the first example of a riboswitch wherein the expression platform binds to a secondary ligand and causes a change in expression and (2) the sequence that binds to the tRNA appears to not share a mechanism of recognizing tRNA with other known T-Box RNAs.

A key point brought up by Reviewer 1 (point 6 in the response letter) is that the SAM-dependent effects on the transcriptional reporter could (and likely are) caused by alternative effects such as stability. I agree with the authors that their revisions address this point. A second and perhaps more important key point brought up by Reviewer 1 (point 12) is that the binding of tRNA to the leader should be supported with better evidence, particularly evidence that this interaction does not occur by spurious base pairing. I agree strongly. Given that this is perhaps the key finding of the paper it is critical that it is correct. The authors have responded to both Reviewers 1 and 2 here. Experiments now included in Figure 4b demonstrate that the binding of tRNA is highly sequence dependent and mutations can abolish binding. However they do not answer one of the other key points – if specificity determinants are not required, why do other cellular tRNAs not bind? Can the authors show that they do not bind to other tRNAs by a similar EMSA assay? Even showing that a few other tRNAs do not bind would be sufficient. This control is lacking and critical to demonstrate specificity. In my opinion once this is demonstrated the paper will be suitable for publication.

Rebuttal letter for NCOMMS-19-17444A

Point by point response to reviewers

Reviewers' comments:

Reviewer #2 (Remarks to the Author):

Second Review for :

Novel variant riboswitch with a sensory and regulatory dual functioning expression platform

Tang et al. have provided a rebuttal which answers many of my concerns (apparently as well as for reviewer 1). I still think that this is a very interesting manuscript with big potential for impact.

However, there are still many minor points that would require revision, as well as a few potentially more important aspects. For this second revision, I have gone through both the rebuttal to my previous comments and to those of reviewer 1.

In order of appearance in the rebuttal and/or text:

1. The following should be added to the amended text in p6, lines 8-14 of the current version: "... or that this effect may be due to indirect effects of SAM on transcription in general."

Response: Thanks for your suggestion. According to your suggestion, the possibility "that this effect may be due to indirect effects of SAM on transcription in general" has been added in the current revised manuscript (P6 line 24-25).

2. Perhaps the biggest comment I still have to make concerns the "new type of t-box" reported in this manuscript. I agree with reviewer 1 that this is a bold claim and that, while it is supported by a few lines of evidence in the first version of the manuscript, further support would be welcome to make sure the proposed model is right for the "big picture" (even if some details, such as the exact structure of P5,P6,P7, might differ from the current prediction). The revised version includes a few such experiments, such as the RNase H probings and changing CCA to GGU (which surprisingly does not change the ability of the tRNA to bind the expression platform), as well as D-loop and T-loop mutants. With all the mutations assayed, the only ones that help define the proposed specificity for the initiation tRNA-fMet are the 5'U and UA just upstream of the CCA, all the other mutations that affect binding to the expression platform are in features common to all tRNAs (D-loop, T-loop, deleted CCA or added 3'C). I noted the possibility of 9 bp between the two RNAs, which actually fits with results of expression platform mutants. The M2 mutation changes AUCC to UAGG, completely abolishing the capacity of this RNA to interact with

tRNA^{fmet}. This sequence could base pair with tRNA, actually, there are 9 consecutive bases that can do Watson-Crick bp with the tRNA first junction and 5' end (CGGGGUGGA of tRNA can bp with UCCACCCCG of stem 7, which also fits with a loop of that structure, so presumably more accessible). While less drastic, the M6 would disrupt one of bp of that interaction, perhaps this would suffice in preventing it. M7, would behave as M2.

The dilemma is to either properly support the “new type of t-box”, which corresponds to an important discovery of a novel sensing non-coding RNA structure which would be a high impact finding, or tune down this finding and simply present this as a potential way by which the riboswitch expression platform may be somewhat affected. For the former, several lines of evidence are already presented, but additional controls could further help dissipate doubts regarding the model. For instance, for Fig6, a negative control where one of the expression platform mutants known not to interact with the tRNA (like 8004/pM2-SD-) is tested in the WT strain or those overexpressing the tRNAs (especially tRNA^{fMet}) would dissipate doubt regarding a potential general effect on translation of overexpressing the initiation tRNA (although, admittedly, the RpoB control seems rather constant). As for in-vitro tests, it is relatively common to include competitors, especially given the fact that at 10uM, non-specific interactions might not be so surprising. Regarding this, in their rebuttal, authors mention that the base pairing is unlikely to be gratuitous because some single point mutations suffice to prevent the interaction, but if the fortuitous interaction happens to occur through base-pairing in that region (or through base-pairing with bases which availability is affected by that region), it could have such an impact as well. In other words, while it does seem like a specific interaction, we cannot completely rule out a relatively non-specific interaction. Another experiment that could help in that regard is determining the K_d, which would be possible with the EMSA assays the authors have used. That being said, given that the range of cellular concentrations of tRNAs, which just so happens to be in the range of ~2 to 10 uM for methionine tRNAs in E. coli

(<https://bionumbers.hms.harvard.edu/files/The%20intracellular%20concentration%20of%20B5M%20of%20tRNA%20isoacceptors%20as%20a%20function%20of%20growth%20rate.pdf>), thus suggesting these concentrations are not so high after all.

Response:

(1) About the claim that SAM-I_{Xcc} is a new type of T-box. According to your comment, we have deleted the claim that SAM-I_{Xcc} is a new type of T-box. All of the description relative to this issue throughout the manuscript have been revised accordingly (page 2, line 16 and line 18; page 15, line 15-16; page 17, line 6 in the revised version).

(2) About the negative control in the original Fig. 6. As you pointed out, RpoB is a meaningful control. We think that this control is sufficient to rule out the possibility that the elevated XC1251(MetA) in the tRNA^{fMet} overexpressing strain is caused by a general effect on translation, although an additional control is useful. Please note that XC1251 elevation only happened in the presence of SAM but did not happen in absence of SAM. If tRNA^{fMet} overexpression could lead to an increase in general translation, XC1251 level should also increase in absence of SAM. To avoid confusion, as shown in Fig. 5B in the current version (the revised Figure of the

original Fig 6B), we used percentage value to compare protein levels in the presence or absence of SAM.

(3) About competitive EMSA. Thanks for your suggestion! According to your suggestion, we performed competitive EMSA experiments using an excess (25-200 fold) of unlabelled wild-type tRNA^{fMet}, a tRNA^{fMet} mutant that lost tRNA^{fMet}-binding ability or the initiator tRNA^{Met1} as a competitor. As expected, the results support the specificity of SAM-I_{Xcc}-tRNA^{fMet} interaction. In addition, according to Reviewer #3's suggestion, we tested the *in vitro* binding of SAM-I_{Xcc} with 19 other tRNAs (the genome of *Xcc* strain 8004 encodes 54 putative tRNAs including the three Met-tRNAs) by EMSA and found that none of the 19 tRNAs could bind to SAM-I_{Xcc}, further demonstrating the specificity. These results have been added into the main text (page 10, lines 14-20) and the experimental data were shown in Supplementary Figures 16 and 17 in the current revised version.

(4) About the RNA concentration used in EMSA. Thanks for providing the information about the intracellular concentration of tRNA in *E. coli*! In our EMSA experiments, the final concentration of tRNA is 1nM, and SAM-I_{Xcc} concentration is 10 μM. Thus, the tRNA concentration used is much lower than that in living cells.

3. P7 line 18 to P8 line 16: While the authors did change several aspects of the text, I am not convinced this section is absolutely necessary in the main text, the effect of the stems on transcription (or even RNA stability) is fairly small if the proposed stems are really involved in such a regulatory mechanism. I would argue that this whole section could be summarized in ~2 sentences briefly describing how deleting each stem individually adds no significant effect on RNA levels, and that two or three of the stems needed to be deleted to have an effect of ~2 fold increase compared to the three stems found in WT.

Response: According to your suggestion, the section (P7, line 18 to P8, line 16 in previous version) has been deleted and the relative result was condensed into a few sentences in the current version (page7, lines 1-11). The previous Fig. 2 has been moved from the main text to Supplementary Information as Supplementary Fig. 4 in the current version.

4. If authors do keep that section (or wish to move it to supplementary material), here are some minor corrections:

P7line21: the decaeson of -> the decrease of

Line 22: To confirme -> To confirm

P8 line 3: lacking -> lack

Line 9: bythe -> by the

Response: The section has been deleted and the relative Fig. 2 has been moved to Supplementary Information as Supplementary Fig. 4.

5. Page9 line 21: Authors mention that GUS activities in absence of SAM "are very similar" in Fig 3c, actually, they are identical. Clearly, each of the constructs was normalized to its "no SAM" condition. So this statement can't be deduced from fig 3c,

we would need un-normalized data to say this. The text needs to be changed and/or data presented differently.

Response: You are right. However, here, “the GUS activities” means the absolute GUS activity value, which was shown in the figure legend of original Fig. 3 (page 30, lines 19-24 in the previous version). To avoid confusion, in the current revised version, the absolute GUS activity value for each of the reporters was removed from the figure legend and directly shown in the figure (Fig. 2C in the current version).

6. Page 9 line 23: *explantation* -> *explanation*

Lin 25: *is existent* -> *exists*

Also on line 25, “Fig. 5b” is mentioned before “fig 4” and suppl fig 16 before many other suppl figures, although this does not cause problems to the reader, it is somewhat unusual not to number figures in order of appearance.

Response: Thanks for pointing out this issue. The figures in the main text and the supplementary information have been renumbered according to the order of appearance in the revised manuscript. The two typos have been corrected (page 9, line 16 and line 18).

7. Page 17 line 10:

The statement “a new type of riboswitch”, while better than the previous version, might still be better if changed the more precise statement: “a riboswitch with a new type of dual sensing capacity”

Response: The statement has been deleted in the current revised version (page 17, line 6).

8. Page 21 line 12: *none-labeled* -> *unlabeled*

Response: The word “none-labeled” has been changed to “unlabeled” in the current revised version (page 21, line 12).

9. Page 30 line 4: *insersion* -> *insertion*

Response: The typo has been corrected in the current revised version (page 5, line 8 in Supplementary Information).

10. *The following comments/correction follow the numbering of the second version of the manuscript at the end of the file.*

Line 299: presumably authors meant GGU instead of UUG.

Line 311: Furtheromre -> Furthermore

Response: You are right. GGU is correct. The two typos have been corrected in the current revised version (page 11, line 8 and line 20).

11. *Regarding the rebuttal for the reference of SAM transporters, while the new reference does refer to bacteria SAM transport, it is a very specific case. The abstract of that reference actually states:*

"We have confirmed the presence of an AdoMet transporter in the rickettsiae which, to our knowledge, is the first bacterial AdoMet transporter identified."

So it actually argues against specific transport of SAM and, browsing through the article, I could not find anything related to non-specific transport of SAM through amino acid permease.

Many important experiments of this manuscript rely on SAM supplementation in the medium, but even if there are no known SAM transporters, some non-specific transport could presumably occur (either of SAM itself or of SAM breakdown products, such as methionine). The authors' results clearly show riboswitch-mediated regulation in presence of SAM, which is completely in accordance with what would be expected if intracellular SAM concentrations increased in these conditions. In other words, this does not jeopardize the whole manuscript, but the statement still remains inappropriate in its current format.

Response: Thank you very much for pointing out this issue. You are right, the reference cited in last revised version describes the identification of a specific transport of SAM, and does not mention about "non-specific transport of SAM through amino acid permease". In fact, the statement "take up SAM by a general amino acid permease" comes from the paper by Kraidlova et al [Characterization of the *Candida albicans* amino acid permease family: Gap2 is the only general amino acid permease and Gap4 is an S-adenosylmethionine (SAM) transporter required for SAM-induced morphogenesis. *mSphere* 1, e00284-00216 (2016).], which we cited in the initial submitted version. Since the paper described a fungal SAM transport, you required us to replace it by a publication regarding bacterial SAM transporter. In the last revised version, we replaced it by the current one, but did not change the statement accordingly.

A literature searching revealed that, only two bacterial SAM transporters have been identified so far, one is from *Rickettsia prowazekii* and the other is from *Chlamydia trachomatis* [Binet, R. et al. Identification and characterization of the *Chlamydia trachomatis* L2 S-adenosylmethionine transporter. *mBio*. 10;2(3):e00051-11 (2011)]. Thus, it is hard to use published data to support the conclusion that *Xcc* can directly uptake SAM from medium. To fill this gap, we tested the growth of the *met* operon inactivation mutant 1201PK2 (a strain we constructed previously), which is unable to synthesize Met and SAM in the presence or absence of extracellular SAM. The results showed that addition of SAM could restore the growth of 1201PK2 in the minimal medium MMX, suggesting the presence of a SAM transporter in *Xcc*. This result has been added in the main text (page 6, lines 8-11 in the current revised manuscript) and the data were shown in Supplementary Fig. 3. The statement "It is known that bacteria can take up SAM directly by a SAM-specific transporter or a general amino acid permease" has been changed to "It is known that bacteria can take up SAM directly by a SAM-specific transporter" and the Binet, R. et al. paper was added as an additional reference in the current revised version (page 6, line 8).

12. Line 152:

for the proposed changes, instead of I would rather suggest:

"... corresponds to SAM-I aptamer..."

Response: According to your suggestion, "...is nearly identical to SAM-I aptamer..." has been changed to "... corresponds to SAM-I aptamer..." in the current revised version (page 8, line 6).

13. To clarify the author's rebuttal, in their response to my concerns with supplementary figure 9 (now figure 2) (p11-12 of the rebuttal pdf), authors' mention that the GUS activity was affected by SAM addition, I assume they meant "was not affected", otherwise I disagree with their interpretation.

Response: Yes, you are right, we meant "was not affected". We are sorry for this mistake.

14. It would be interesting to mention somewhere in the manuscript that Xcc encodes two homoserine O-acetyltransferases, as this may be meaningful regarding the biology of the regulation of this gene.

Response: According to your suggestion, a few sentences describing the other homoserine O-acetyltransferase-encoding gene (XC1889) have been added in the current revised version (page 5 lines 14-20).

15. Regarding the addition of an alignment, while Xhang et al. 2018 present an alignment of the UTR, it was not described as an RNA structure in this manuscript, as such it would still be beneficial to include an alignment to allow readers to judge the conservation of the motif with proper format to evaluate the structures described (at least in supplementary material).

As a helper:

```

Query 1          ACGCGAGCTCCC CGCGAAGCTCGATGGCCGATCCACC..CCGGATATCGCCATG 51
CP000050.1    1531183   ACGCGAGCTCCC CGCGAAGCTCGATGGCCGATCCACC..CCGGATATCGCCATG 1531233
CP016830.1    1274832   ACGCGAGCTCCC CGCGAAGCTCGATGGCCGATCCACC..CTGGATATCGCCATG 1274882
CP018728.1    4131786   ACGCGAGCTCC-GCGAAGCTCGATGGCCGATCCACC..CCGGATACCGCCATG 4131737
CP019515.1    3062879   ACGCGAGCTCC-GCGAAGCTCGATGGCCGATCCACC..CTGGATACCGCCATG 3062830
CP020971.1    4621205   ACGCGAGCTCC-GCGAAGCTCGATGGCCGATCCACC..CTGGATACCGCCATG 4621254
CP034653.1    1907635   ACGCGAGCTCC-GCGAAGCTCGATGGCCGATCCACC..CTGGATACCGCCATG 1907684
CP020987.1    934400    ACGCGAGCTCC-GCGAAGCTCGATGGCCGATCCACC..CTGGATACCGCCATG 934351
CP013004.1    1576411   ACGCGAGCTCC-GCGAAGCTCGATGGCCGATCCACC..CTGGATACCGCCATG 1576460
CP022270.1    3461809   ACGCGAGCTCC-GCGAAGCTCGATGGCCGATCCACC..CTGGATACCGCCATG 3461760
CP013679.1    1768456   ACGCGAGCTCC-GCGAAGCTCGATGGCCGATCCACC..CTGGATACCGCCATG 1768407
CP019797.1    2553242           CCCGCGAAGCTCGATGGCTGTCCTTA..CCGGATATCGCCATG 2553201
CP033586.1    1002918   ACGCGAGCTCCC CGCGAAGCTCGATGGCCGATCCACC..CCGGATACCGCCATG 1002970
LS483406.1    3498410   ACGCGAGCTCCC CGCGAAGCTCGATGGCCGATCCACC..CCGGATACCGCCATG 3498358
    . . . . .  "*"*****"  ""*****"  "* *    * * * *   *****

```

Colors at the top highlight the base pairing regions P6 (green), P7a (yellow) and P7b (turquoise). Positions that vary in the alignment are also colored (red=changes that disrupt base pairs; grey= "silent mutations, or gap"; blue=mutations that still permit base pairing; pink=mutations that extend base pairs). As seen from this, there is very little support for the proposed structure model with regards to co-variation. However, given that the sequence is indeed very conserved (and many other sequences from other strains and species identical to these ones), this is not so surprising, but does require mutational analysis to confirm the structure and important elements of it, which the authors have done at least in part.

Response: Thank you very much for your suggestion and making an alignment for us. According to your suggestion and guidance, an alignment showing the sequences and secondary structure conservation of SAM-I_{Xcc} in *Xanthomonas* has been added in the current revised manuscript as Supplementary Fig. 24.

The last three sequences (LS483406.1, CP033586.1 and CP019797.1) shown in the alignment you provided are not from *Xanthomonas* strains but from *Stenotrophomonas maltophilia* strain SVIA2, *Stenotrophomonas maltophilia* strain NCTC10498 and *Stenotrophomonas acidaminiphila* strain SUNE0, respectively. In the alignment (Supplementary Fig. 24) of SAM-I_{Xcc} homologues in *Xanthomonas* strains, only a few single nucleotide changes are found in the base pairing region, and it seems that these changes should not disrupt the stem. According to your suggestion, we did EMSA experiments to detect whether mutations in these nucleotides in SAM-I_{Xcc} affect its tRNA^{fMet}-binding ability. The results showed that none of these mutations affected the binding of SAM-I_{Xcc} to tRNA^{fMet}. The results were described in the last paragraph of the Discussion section (page 17, lines 14-17) in the current revised manuscript and the data were shown in Supplementary Fig. 25.

16. *Regarding the figure taken from another publication, I suggest the authors properly review the copyrights and each journal details regarding permissions required for inclusion of even portions of a figure in another article. Also, while the correct journals are cited, authors should add something such as “adapted from” or “taken from” in the legend of the figures borrowed from other articles. Looking back at some figures, I also noted that suppl fig 8 (a) and (b) come from the cited reference and that (d) and (e) come from the other corresponding cited reference. So, even though this was likely included in good faith, some verifications or permissions are required.*

Response: Thanks for your suggestion. We have sent our request for permission to use the figures to the journals and have revised the figure legends by adding a statement of “taken from...” (please see the legends of Supplementary Figures 10 and 14 in the current revised version).

Reviewer #3:

*In this manuscript, Tang et al. report a novel riboswitch from *Xanthomonas campestris* (Xcc). This riboswitch recognizes S-adenosyl methionine (SAM), and in doing so appears to function similar to a classical SAM riboswitch – by changing conformation in the presence of SAM, resulting in altered gene expression. Indeed, the SAM-IXcc switch was identified in part through a homology search. However, this riboswitch has the unexpected and novel activity of also binding to and responding to uncharged initiator Met tRNA. The authors demonstrate ligand-dependent conformational changes (in response to SAM) by in-line probing. These effects while convincing are somewhat subtle and temperature-dependent: the authors posit that this is due to high GC-content or due to the modest Kd of SAM, though it could also be that in this unusual system the aptamer behaves somewhat differently in isolation than in the context of the expression platform. Further, the authors report in an expression system that (1) the effects of the switch occur at the translational level, (2) three stem loops in the expression platform are critical to activity (one of which*

occludes the Shine-Delgarno sequence), (3) the Met tRNA binds to one of these hairpins, and in doing so influences translation activity.

This paper has undergone extensive revision in response to a previous round of reviews. The revised manuscript is clearly written, and the authors have responded to all reviewer comments. I agree strongly with the previous reviewers that the key points in this manuscript are related to the tRNA-sensing aspects of this riboswitch. These discoveries are novel and warrant publication in Nature Communications for two reasons – (1) this is the first example of a riboswitch wherein the expression platform binds to a secondary ligand and causes a change in expression and (2) the sequence that binds to the tRNA appears to not share a mechanism of recognizing tRNA with other known T-Box RNAs.

A key point brought up by Reviewer 1 (point 6 in the response letter) is that the SAM-dependent effects on the transcriptional reporter could (and likely are) caused by alternative effects such as stability. I agree with the authors that their revisions address this point. A second and perhaps more important key point brought up by Reviewer 1 (point 12) is that the binding of tRNA to the leader should be supported with better evidence, particularly evidence that this interaction does not occur by spurious base pairing. I agree strongly. Given that this is perhaps the key finding of the paper it is critical that it is correct. The authors have responded to both Reviewers 1 and 2 here. Experiments now included in Figure 4b demonstrate that the binding of tRNA is highly sequence dependent and mutations can abolish binding. However they do not answer one of the other key points – if specificity determinants are not required, why do other cellular tRNAs not bind? Can the authors show that they do not bind to other tRNAs by a similar EMSA assay? Even showing that a few other tRNAs do not bind would be sufficient. This control is lacking and critical to demonstrate specificity. In my opinion once this is demonstrated the paper will be suitable for publication.

Response: Thank you very much for your suggestion. According to your suggestion, we tested the *in vitro* binding of SAM-I_{Xcc} with 19 other tRNAs (the genome of Xcc strain 8004 was predicted to encode a total of 54 tRNAs including the three Met-tRNAs) by EMSA and the result showed that none of the 19 tRNAs tested could bind with SAM-I_{Xcc}. This result supports that SAM-I_{Xcc}-tRNA^{fMet} interaction is specific. Furthermore, the binding specificity between SAM-I_{Xcc} and tRNA^{fMet} was further confirmed by competitive EMSA assay (suggested by Reviewer #2) using an excess (25-200 fold) of unlabelled wild-type tRNA^{fMet}, a tRNA^{fMet} mutant that lost tRNA^{fMet}-binding ability or the initiator tRNA^{Met1} as competitors. These results have been added into the main text (page 10 lines 14-20) and the data were shown in Supplementary Figures 16 and 17, respectively, in the current revised version.

REVIEWERS' COMMENTS:

Reviewer #2 (Remarks to the Author):

Third Review for :

Novel variant riboswitch with a sensory and regulatory dual functioning expression platform

Tang et al. have provided a rebuttal which answers many of my concerns (apparently as well as for reviewer #3). In addition to many added information, like the RNA structure alignments, a significant amount of work was done to get results to further support the important "tRNA-sensing" capacity of this non-coding RNA and it is to be expected that this will become a highly cited article as the second example of such a tRNA-sensor (after T-box, discovered as a tRNA sensor in 1993). Following my suggestion, which was intended to tune down the authors claims regarding the tRNA-sensing capability of this RNA structure, the authors changed "new T-box" to "tRNA-sensing". Eventhough now I do not think it is necessary to make that change, given that the authors provided additional evidence, mainly with the many additional tRNAs tested by EMSA, I would keep this new revised version, as it avoids confusion with T-boxes, thus emphasizing how different this RNA is from the known T-box structure.

I have a few very minor comments, these changes can be made without a requirement for further revision on my part. In order of appearance in the rebuttal and/or text:

All the 3' and 5' should be changed to have a "prime" symbol, not an apostrophe

Line 34: identify an SAM-I -> identify a SAM-I

Point #16 in rebuttal:

The authors mention that permission to use some figures was requested, before final publication, authors (and Nature Communication) should make sure proper permission is obtained for the corresponding supplementary figures.

Overall, I think this is a very interesting finding, with a lot of high quality data, and that it deserves to be published to get a maximum of readers.

Reviewer #3 (Remarks to the Author):

In my opinion the authors have addressed my concerns and it looks like they have addressed the concerns of the other reviewers as well. I recommend the manuscript be accepted.

REVIEWERS' COMMENTS:

Reviewer #2 (Remarks to the Author):

Third Review for :

Novel variant riboswitch with a sensory and regulatory dual functioning expression platform Tang et al. have provided a rebuttal which answers many of my concerns (apparently as well as for reviewer #3). In addition to many added information, like the RNA structure alignments, a significant amount of work was done to get results to further support the important "tRNA-sensing" capacity of this non-coding RNA and it is to be expected that this will become a highly cited article as the second example of such a tRNA-sensor (after T-box, discovered as a tRNA sensor in 1993). Following my suggestion, which was intended to tune down the authors claims regarding the tRNA-sensing capability of this RNA structure, the authors changed "new T-box" to "tRNA-sensing". Eventhough now I do not think it is necessary to make that change, given that the authors provided additional evidence, mainly with the many additional tRNAs tested by EMSA, I would keep this new revised version, as it avoids confusion with T-boxes, thus emphasizing how different this RNA is from the known T-box structure.

I have a few very minor comments, these changes can be made without a requirement for further revision on my part. In order of appearance in the rebuttal and/or text:

All the 3' and 5' should be changed to have a "prime" symbol, not an apostrophe

Response: The 3' and 5' have been changed from an apostrophe to a "prime" symbol in the current version of the submission.

Line 34: identify an SAM-I -> identify a SAM-I

Response: "an SAM-I" has been changed to "a SAM-I" in the current version of the submission.

Point #16 in rebuttal:

The authors mention that permission to use some figures was requested, before final publication, authors (and Nature Communication) should make sure proper permission is obtained for the corresponding supplementary figures.

Response: We have already obtained the permission from the copyright holders of the related images (see the attachments).

Overall, I think this is a very interesting finding, with a lot of high quality data, and that it deserves to be published to get a maximum of readers.

Reviewer #3 (Remarks to the Author):

In my opinion the authors have addressed my concerns and it looks like they have addressed the concerns of the other reviewers as well. I recommend the manuscript be accepted.